# The effects of 0.9% saline versus Plasma-Lyte 148 on renal function as assessed by creatinine concentration in patients undergoing major surgery: A single-centre double-blinded cluster crossover trial

Laurence Weinberg[1,2]*, Michael Hua-Gen Li[2], Leonid Churilov[3], Christopher Macgregor[1], Kent Garrett[4], Jade Eyles[4], Rinaldo Bellomo[5]

1 Department of Anaesthesia, Austin Health, Heidelberg, Victoria, Australia, 2 Department of Surgery, The University of Melbourne, Austin Health, Heidelberg, Victoria, Australia, 3 Department of Medicine (Austin Health) and Melbourne Brain Centre at Royal Melbourne Hospital, The University of Melbourne, Heidelberg, Victoria, Australia, 4 Department of Pharmacy, Austin Health, Heidelberg, Victoria, Australia, 5 Department of Intensive Care, Austin Hospital, Heidelberg, Victoria, Australia

* laurence.weinberg@austin.org.au

## Abstract

### Objectives

Saline and Plasma-Lyte have different physiochemical contents; consequently, they may differently affect patients' renal function. We compared the effects of fluid therapy with 0.9% saline and with Plasma-Lyte 148 on renal function as assessed by creatinine concentration among patients undergoing major surgery.

### Methods

We conducted a prospective, double-blinded cluster crossover trial comparing the effects of the two fluids on major surgery patients. The primary aim was to establish the pilot feasibility, safety and preliminary efficacy evidence base for a large interventional trial to establish whether saline or Plasma-Lyte is the preferred crystalloid fluid for managing major surgery patients. The primary efficacy outcome was the proportion of patients with changes in renal function as assessed by creatinine concentration during their index hospital admission. We used changes in creatinine to define acute kidney injury (AKI) according to the RIFLE criteria.

### Results

The study was feasible with 100% patient and clinician acceptance. There were no deviations from the trial protocol. After screening, we allocated 602 patients to saline and 458 to Plasma-Lyte. The median (IQR) volume of intraoperative fluid received was 2000 mL (1000:2000) in both groups. Forty-nine saline patients (8.1%) and 49 Plasma-Lyte patients (10.7%) developed a postoperative AKI (adjusted incidence rate ratio [aIRR]: 1.34; 95% CI:

**Data Availability Statement:** All relevant data are within the manuscript and its Supporting Information files.

**Funding:** Baxter Healthcare assisted with the manufacturing and compounding of the trial fluids. Baxter Healthcare provided the trial fluid in indistinguishable 1000 mL bags arbitrarily labelled 'Surgilyte A' (blinded Plasma-Lyte 148) or 'Surgilyte B' (blinded saline). The funders had no role in study design, data collection and analysis, decision to publish, or preparation of the manuscript.

**Competing interests:** I have read the journal's policy and the authors of this manuscript have the following competing interests: Prof Rinaldo Bellomo and A/Prof Laurence Weinberg have received honoraria of <US$5000 from Baxter Healthcare for consulting activities. The Australian and New Zealand Intensive Care Research Centre and the Departments of Intensive Care and Anaesthesia at Austin Health have received research grants from Baxter Healthcare. All aspects of the study design, execution, data collection, and analysis have been conducted independently of Baxter Health or any other industry. This does not alter our adherence to all PLOS ONE policies on sharing data and materials.

0.93–1.95; p = 0.120). No differences were observed in the development of postoperative complications (aIRR: 0.98; 95% CI: 0.89–1.08) or the severity of the worst complication (aIRR: 1.00; 95% CI: 0.78–1.30). The median (IQR) length of hospital stay was six days (3:11) for the saline group and five days (3:10) for the Plasma-Lyte group (aIRR: 0.85; 95% CI: 0.73–0.98). There were no serious adverse events relating to the trial fluids, nor were there fluid crossover or contamination events.

## Conclusions

The study design was feasible to support a future follow-up larger clinical trial. Patients treated with saline did not demonstrate an increased incidence of postoperative AKI (defined as changes in creatinine) compared to those treated with Plasma-Lyte. Our findings imply that clinicians can reasonably use either solution intraoperatively for adult patients undergoing major surgery.

## Trial registration

Registry: Australian New Zealand Clinical Trials Registry; ACTRN12613001042730; URL: https://www.anzctr.org.au/Trial/Registration/TrialReview.aspx?id=364988.

## Introduction

Postoperative acute kidney injury (AKI) occurs in up to 20% of hospital patients, accounts for 18–47% of in-hospital AKI and is associated with increased length of stay, morbidity and mortality [1]. While liberal intravenous (IV) fluid administration reportedly reduces AKI in patients receiving major abdominal surgery during the perioperative period [2], controversy remains regarding the optimal fluid for use in this setting. For example, 0.9% sodium chloride solution (saline) remains in use for IV fluid therapy in the United Kingdom [3] and United States [4], despite the fact that numerous studies in the past decade have examined the effects of hyperchloraemic solutions on renal function [5–11], and most have suggested that balanced crystalloid solutions with lower chloride concentrations may have less nephrotoxic effects. Hyperchloraemia has been reported to be associated with chloride-induced renal vasoconstriction, however, more recent studies [12–14] have reported that major morbidity, including AKI, is comparable between patients treated with saline and those treated with lactated balanced crystalloids. Other studies have yielded conflicting results [15–18].

With over 230 million major surgical procedures occurring annually worldwide [19], preventing perioperative AKI remains a significant public health issue. To date, no prospective studies have examined the effectiveness of saline versus Plasma-Lyte 148 (Plasma-Lyte) for fluid therapy in patients undergoing major surgery. Accordingly, we designed and conducted a prospective, double-blinded cluster crossover trial to determine the comparative effectiveness of crystalloid fluid therapy using both fluids in a heterogeneous population of patients undergoing major elective or emergency surgery. The primary efficacy outcome was the proportion of patients with changes in renal function as assessed by creatinine concentration during their index hospital admission. We used changes in creatinine to define acute kidney injury (AKI) according to the RIFLE criteria.

## Methods

### Ethics

The study was approved by the Human Research Ethics Committee at Austin Health (HREC/13/Austin/161) and prospectively registered with the Australian New Zealand Clinical Trials Registry (ACTRN12613001042730). Similar to other studies [12], because this investigation involved the systematic evaluation and comparative effectiveness of two commonly applied treatments, the process of opt-out consent for all patients was approved by the ethics committee. Given that this study was a pragmatic comparative effectiveness study and that participation in the research posed no more than minimal increase in risk to individuals than what they would be exposed to if they were not in the study, the ethics committee granted waiver of participant consent. A predefined statistical analysis plan was reported and published before study completion [20]. The ethically approved trial protocol is presented as a supplementary file (see **S1 Protocol**).

The study was an investigator-initiated, prospective, double-blinded cluster crossover trial. It was conducted within the Saline v. Plasma-Lyte for IV Fluid Therapy (SPLIT) research program, a binational, multidisciplinary investigation of the comparative effectiveness of IV saline and Plasma-Lyte in fluid therapy [20]. The primary aim was to establish the pilot feasibility, safety and preliminary efficacy evidence base for a large interventional trial to establish whether saline or Plasma-Lyte is the preferred crystalloid fluid for managing major surgery patients.

Patients aged 18 years or older undergoing major surgery in our institution were eligible for inclusion. Major surgery was defined as any procedure lasting more than two hours, requiring surgical incision and necessitating at least one night of postoperative stay. Patients undergoing renal and liver transplantation were excluded (due to surgery-specific fluid protocols for such procedures), as were those with raised intracranial pressure or end-stage renal failure [21] (eGFR $< 15$ mL/min$^{-1}$ for three or more months). We also excluded patients not expected to survive their index hospital admission (American Society of Anesthesiology Class 5). Patients requiring a second operation during the trial period who initially received the study fluid, still received the study fluid in their re-operation. They were included in the intention-to-treat population from their initial surgery, however were excluded from the study for their reoperation.

Similar to our previous studies [12], we used Plasma-Lyte as a comparative fluid to saline due to its physiological profile, which more closely resembles human plasma (see **S1 Table**). Additionally, Hartmann's solution's calcium content renders it incompatible with blood products preserved in citrate-based anticoagulation solutions [22].

Between 29$^{th}$ October 2014 and 15$^{th}$ February 2015 in two alternating six-week periods, all eligible patients were assigned to blinded saline or blinded Plasma-Lyte. Unlike a conventional randomised clinical trial, "clusters" of individuals, rather than individual patients were allocated to receive either blinded saline or Plasma-Lyte over two consecutive predefined 6-week periods. Three weeks before commencement, blinded Plasma-Lyte was introduced into all operating theatres, the intensive care unit (ICU) and surgical wards. Participants then received Plasma-Lyte for a six-week period, followed by a three-week wash-out period, allowing patients exposed to the intervention intraoperatively to continue to receive their assigned fluid. Next, after a three-week wash-in period, participants received saline for six weeks, followed by a three-week wash out. The total study duration was 24 weeks. The trial design is presented in Fig 1. Fluid was compounded by Baxter Healthcare in indistinguishable 1000 mL bags arbitrarily labelled 'Surgilyte A' (blinded Plasma-Lyte 148) or 'Surgilyte B' (blinded saline) (see **S1 Fig**). Investigators, clinicians, nurses and patients were blinded to fluid allocation

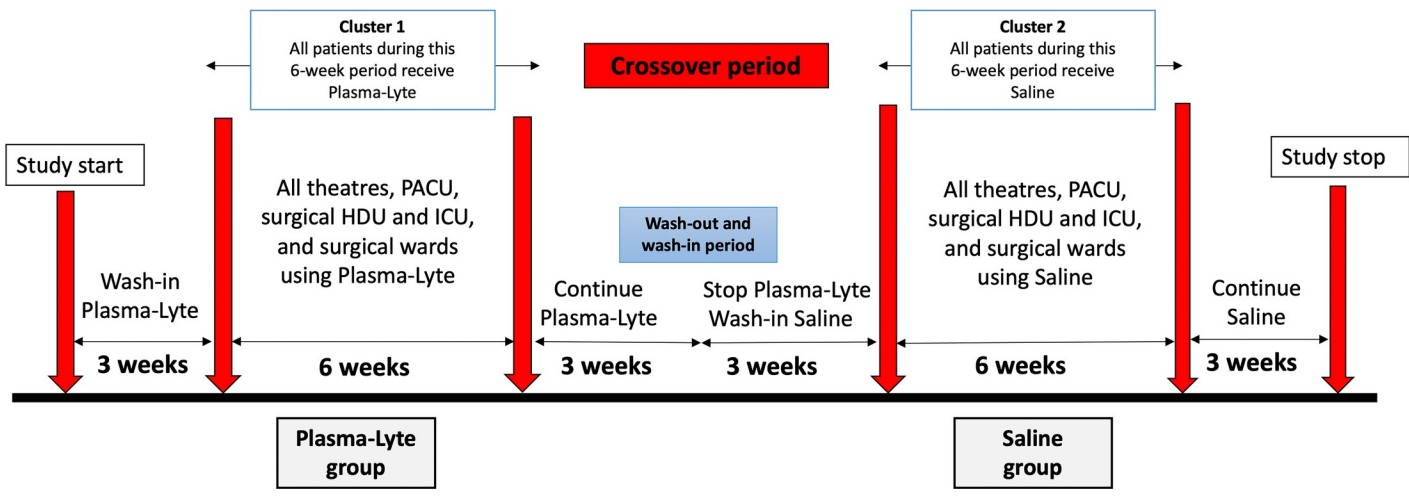

**Fig 1. Cluster crossover design of Saline vs Plasma-Lyte study.**

throughout the study. The use of other fluids or therapies was not restricted. Use of colloids, blood products or other non-crystalloid fluid was at the discretion of the treating clinicians, who also determined the rate and frequency of fluid administration.

Two dedicated pharmacists (KG, JE) were responsible for fluid allocation and supply to theatres, the ICU and the ward. The pharmacy department oversaw the logistical and operational ordering of fluids from Baxter Healthcare. Fluids were stored in a dedicated pharmacy warehouse located within the hospital, and the pharmacists co-ordinated the continuous and timely supply of fluids across the recruiting areas.

Feasibility outcomes included patient and clinician acceptance, recruitment rate, reasons for exclusion, logistical or operational feasibility with regards to fluid storage and delivery to the clinical areas where the trial was undertaken, deviations from the trial protocol, unblinding rate, and number of patients who fulfilled eligibility who did not have renal function measured as part of routine clinical care. The primary efficacy outcome was the proportion of patients with changes in renal function as assessed by creatinine concentration during their index hospital admission. We used changes in creatinine to define acute kidney injury (AKI) according to the RIFLE criteria. AKI was then defined by creatinine levels and assessed according to the risk, injury, failure, loss of kidney function and end-stage renal failure (RIFLE) criteria during the index hospital admission. RIFLE evaluations are based on changes in serum creatinine threshold or estimated glomerular filtration rate (eGFR) and urinary output [23]. Intravenous fluid volume was collected up to 72 hours postoperatively. The urinary output component was not used, since urine output is typically not monitored in ward patients without urinary catheters in situ. Renal function tests were performed at the discretion of the treating clinical unit and only if there was a specific clinical indication to measure renal function in the perioperative setting. Patients on whom such tests were not performed were considered as not having developed an AKI. Safety outcomes included fluid administration errors, unintentional fluid discontinuation, and crossover or contamination events.

Secondary efficacy outcomes included the proportion of patients with AKI defined by creatinine levels and assessed according to the Kidney Disease Improving Global Outcomes

classification [24], the proportion of patients who developed any postoperative complication during their index hospital admission, complication severity, the total number of complications defined by the Classification of Hospital Acquired Diagnoses system [25] and the hospital length of stay.

Outcomes were examined within six predefined patient subgroups known to affect postoperative AKI: surgical urgency (emergency or elective surgery), surgery type (cardiothoracic, major abdominal, major orthopaedic or vascular), patient age ($\geq 70$ or $< 70$ years), preoperative creatinine ($\geq 120$ or $< 120$ umol.L$^{-1}$), preoperative eGFR ($\geq 90$, 60–90 or $< 60$) and Charlson comorbidity index (0–2, 3–4 or $> 4$).

Guided by the European Perioperative Clinical Outcome and Clavien Dindo classification definitions [26, 27], complications were defined as any deviation from the normal postoperative course. Reporting of complications were evaluated by two authors independently (MHL, CM) by undertaking an in-depth review of each patient's clinical records. In the case of disagreement, the case was presented to two other authors (LW, RB) to reach consensus. Length of stay was the period from surgery completion to discharge, excluding days in the hospital-in-the-home unit. Readmission was defined as unplanned readmission within 30 days of discharge. Mortality was considered only when it occurred during the index hospital admission. Preoperative data points included demographics, preoperative biochemistry and Charlson comorbidity index. Intraoperative measurements included duration of surgery, surgical urgency, type of surgery and volume of fluid administration.

## Statistical analyses

Like the other fluid intervention trial designs [12], our study was performed to establish the feasibility of using a crossover design to investigate fluid therapy in the perioperative setting. All SPLIT studies were scheduled to run for a specific period and had no fixed recruitment number [20]. We conducted all analyses on an intention-to-treat basis, in accordance with the predefined statistical analysis plan. Patients who did not undergo renal function tests were included in the final analyses as not having developed an AKI.

We performed statistical analysis using commercial statistical software STATA/IC v.14 (StataCorp, College Station, TX, USA). Unless otherwise stated, results were summarised as either a median (interquartile range [IQR]) or counts and proportions. Comparisons between categorical variables were made using chi-square and Fisher's exact tests, while continuous variables were compared using the Mann–Whitney U test.

We used a covariate-adjusted modified Poisson regression with robust error estimation [28] to compare the trial treatments in analyses of the prespecified primary outcome, dichotomous secondary efficacy outcomes and dichotomous safety outcomes. The primary outcome was adjusted for surgery type, emergency status and eGFR level (trichotomised as $< 60$, 60–90, $> 90$). Treatment effects for all above outcomes are reported as adjusted incidence rate ratios (aIRRs) with 95% confidence intervals (95% CIs). We conducted time-to-event analyses using Cox proportional hazard regression models, and compared non-dichotomous outcomes using Poisson or negative binomial regression models, all adjusted for the same set of covariates, with treatment effects reported as adjusted hazard ratios with 95% CIs. The threshold for statistical significance was a P value of 0.05. All secondary and exploratory endpoints are reported as point estimates of treatment effects with 95% CIs [29]. We report both covariate-adjusted and unadjusted outcomes.

We used forest plots to present the primary and key secondary outcomes regarding the consistency of a treatment effect across the subgroups. Individual P values are presented to compare the patients' baseline characteristics due to the cluster-randomised nature of the study.

## Results and discussion

In total, 5646 patients were screened over the duration of the two predefined cluster periods. During the first 6-week cluster period, 2933 patients were allocated to receive saline; during the second 6-week cluster period, 2713 patients were allocated to receive Plasma-Lyte. After exclusions, 1060 patients fulfilled the inclusion criteria—602 patients were allocated to the saline group and 458 to the Plasma-Lyte group. The CONSORT diagram is presented in Fig 2.

### Feasibility outcomes

Of the 5646 patients screened, there were no patients who opted out from the study. The most common reason for exclusion was a surgical duration of less than 2 hours, which occurred in 77.5% of patients that were allocated to saline and 81.5% of patients allocated to Plasma-Lyte.

The baseline characteristics and key process of care aspects were similar between the groups, with the exception of male sex (59.1% v. 51.5%; p < 0.01) and emergency surgery (42.7% v. 35.2%, p = 0.01), both more common in the saline group (see Table 1). The median (IQR) volume of intraoperative fluid received was 2000 mL (range 1000 to 2000) in both groups. There were no logistical difficulties with fluid storage and no operational errors with the pharmacy department overseeing the delivery and supply of the fluids to theatre, ICU and the ward. There were no deviations from the trial protocol and the study drug was continued in 100% of patients in the theatre, the ICU and the ward. There were no anaesthetists, surgeons, intensivists or ward clinicians who refused to include their patients in the study. Given that the fluids were blinded and compounded in indistinguishable 1000 mL bags (see S1 Fig), there was no inadvertent unblinding of the study drug.

Overall, there were 113 patients (10.6%) who were included in the trial who did not have their renal function measured as part of routine standard of care. The proportion of patients in

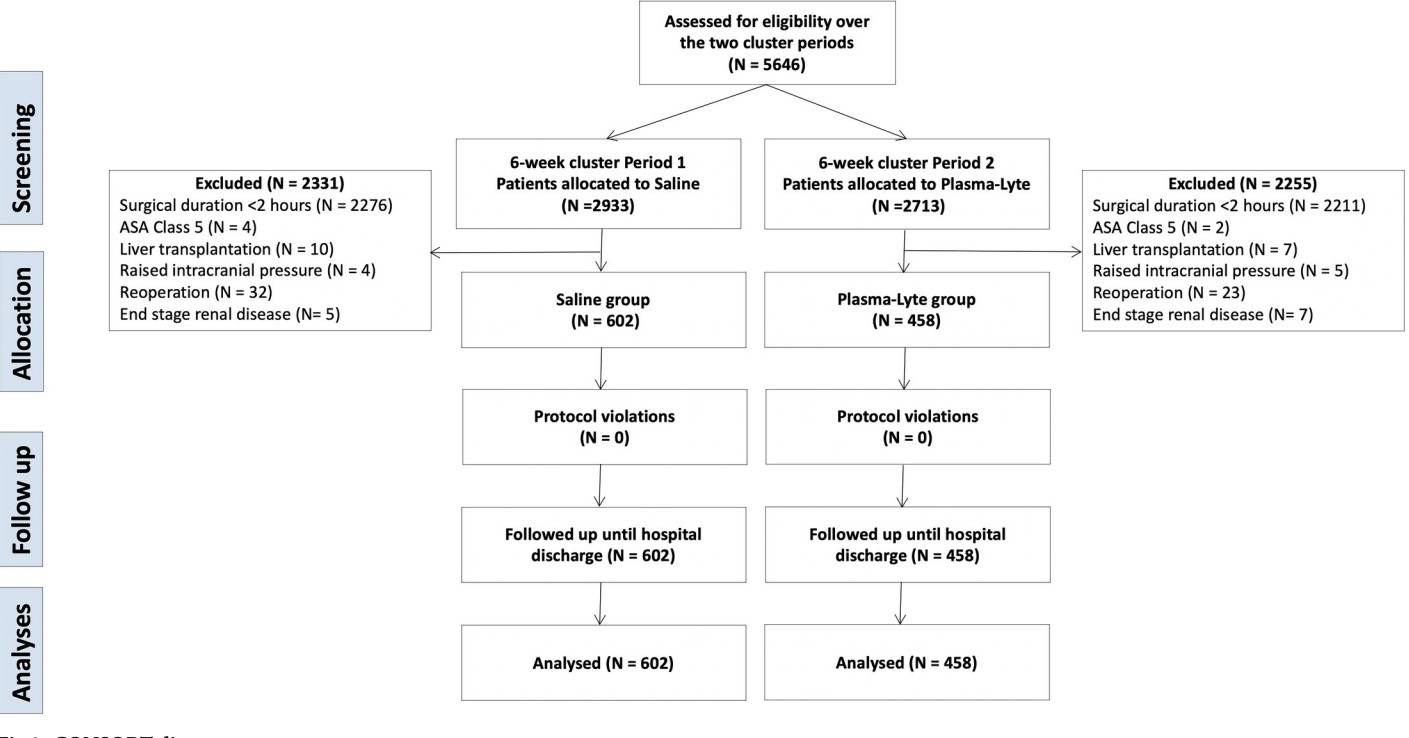

**Fig 2. CONSORT diagram.**

each group were similar (10.5% of patients receiving saline vs. 10.9% of patients receiving Plasma-Lyte, p = 0.841). The differences in the perioperative variables between patients with measured and unmeasured renal function tests are summarised in the supplementary file (S2 Table). Patients with incomplete renal function were younger, with fewer comorbidities, shorter operations and fewer emergency surgeries.

## Effectiveness outcomes

These key outcomes are presented in Table 2. Forty-nine (8.1%) saline patients and 49 (10.7%) Plasma-Lyte patients developed a postoperative AKI assessed by creatinine concentration (aIRR: 1.34; 95% CI 0.93 to 1.95, p = 0.120). Of those saline patients who developed an AKI, 38 (77.6%) were classified as RIFLE class R, three (6.1%) as class I and eight (16.3%) as class F. Of those Plasma-Lyte patients who developed an AKI, 38 (77.6%) were classified as class R, four (8.1%) as class I and seven (14.3%) as class F. No patient in either group was classified as RIFLE class L or E.

No differences were observed between the groups at risk of developing a postoperative complication (aIRR: 0.98; 95% CI 0.89 to 1.08). The severity of the worst complication was also similar between the groups (aIRR: 1.00; 95% CI 0.78 to 1.30). Saline-treated patients developed higher median (IQR) peak serum chloride levels (103 [100 to 106] v. 101 [98 to 103] mmol/L) and lower median (IQR) trough serum bicarbonate levels (23 [21 to 25] v. 25 [22 to 27] mmol/L) on the first postoperative day. The preoperative and postoperative changes in chloride levels are presented as violin plots in Fig 3. Plasma-Lyte treated patients had shorter hospital stays (aIRR: 0.85; 95% CI 0.73 to 0.98). There was no mortality in either group. The forest plots presented in Fig 4 summarise the treatment effects in the patient subgroups.

**Table 1. Perioperative variables in patients receiving Saline and Plasma-Lyte.** Data is median (interquartile range) or number (proportion).

| | | Saline (n = 602) | Plasma-Lyte (n = 458) | p-value |
|---|---|---|---|---|
| **Preoperative factors** | | | | |
| Age, years | | 63 (47–74) | 62 (46–74) | 0.621 |
| Male | | 356 (59.1) | 236 (51.5) | 0.008 |
| Serum creatinine, μmol/L | | 77 (65–94) | 76 (65–93) | 0.792 |
| Serum eGFR, mL/min | | 85 (64–91) | 84 (65–91) | 0.559 |
| Serum chloride, mmol/L | | 101 (98–103) | 100 (98–103) | 0.868 |
| Serum bicarbonate, mmol/L | | 26 (24–28) | 26 (24–28) | 0.168 |
| Serum anaemia[a] | | 184 (31.9) | 127 (29.9) | 0.491 |
| Serum haemoglobin, g/L | | 134 (121–146) | 135 (121–146) | 0.522 |
| Preoperative Charlson Comorbidity Index | | 1 (0–2) | 0 (0–2) | 0.241 |
| **Operative factors** | | | | |
| Duration of surgery, mins | | 200 (151–289) | 197 (153–282) | 0.853 |
| Emergency surgery | | 257 (42.7) | 161 (35.2) | 0.013 |
| Type of surgery (%) | Cardiothoracic | 14.8 | 17.1 | |
| | Major abdominal | 13.6 | 15.5 | |
| | Major orthopaedic | 23.4 | 21.6 | |
| | Vascular | 8.1 | 8.0 | 0.719 |
| | Other [b] | 40.3 | 38.0 | |

eGFR, estimated glomerular filtration rate.

[a]WHO classification [24].

[b]Urological, head and neck, maxillary facial, neurosurgery.

**Table 2. The Incidence Rate Ratios (IRR) and Hazard ratios of outcomes for patients receiving Plasma-Lyte vs Saline after surgery.**

|  | Saline n = 602 | PlasmaLyte n = 458 | Adjusted[a] Effect Size (95%CI) | Unadjusted Effect Size (95% CI) |
|---|---|---|---|---|
| **Primary Outcome** | | | | |
| Patients with RIFLE criteria for AKI[b] | 49 (8.1%) | 49 (10.7%) | IRR 1.34 (0.93 to 1.95) p = 0.120 | IRR 1.31 (0.90 to 1.92) p = 0.155 |
| **Secondary Outcomes** | | | | |
| Patients with RIFLE criteria for AKI[c] | 49 (8.1%) | 49 (10.7%) | HR 1.40 (0.93 to 2.11) | HR 1.39 (0.93 to 2.08) |
| Patients with KDIGO criteria AKI[b] | 61 (10.1%) | 59 (12.9%) | IRR 1.29 (0.93 to 1.80) | IRR 1.27 (0.91 to 1.78) |
| Patients with KDIGO criteria AKI[c] | 61 (10.1%) | 59 (12.9%) | HR 1.37 (0.91 to 2.06) | HR 1.35 (0.91 to 2.01) |
| Patients with a Clavien Dindo complication[b] | 368 (61.1%) | 276 (60.3%) | IRR 0.98 (0.89 to 1.08) | IRR 0.99 (0.89 to 1.46) |
| Worst Clavien Dindo score per patient[b] | N/A | N/A | IRR 1.00 (0.78 to 1.30) | IRR 1.11 (0.85 to 1.46) |
| Number of CHADx complications per patient[b] (median [IQR]) | 1 (0:4) | 1 (0:3) | IRR 1.01 (0.88 to 1.16) | IRR 1.02 (0.89 to 1.18) |
| Length of hospital stay in days[b] (median [IQR]) | 6 (3:11) | 5 (3:10) | IRR 0.85 (0.73 to 0.98) | IRR 0.86 (0.74 to 1.01) |

[a]Adjusted for emergency surgery, surgery type and preoperative estimated glomerular filtration rate.

[b]IRR: Incidence rate ratio for Plasma-Lyte with Saline as the comparator.

[c]HR: Hazard ratio for Plasma-Lyte with Saline as the comparator.

CHADx: Classification of Hospital Acquired Complications, KDIGO: Kidney Disease Improving Global Outcomes, RIFLE: Risk, Injury, Failure, Loss, End-stage renal failure.

## Safety outcomes

There were no study fluid administration errors, and the study fluid was not intentionally or inadvertently discontinued in any patient. There were no serious adverse events related to the specific type of study fluid received. Due to the study design having two distinct and predefined wash-in and wash-out periods, with dedicated pharmacy support for fluid storage and delivery to the clinical areas, there were no crossover or contamination events.

## Key findings

We conducted a double-blinded, cluster crossover study to compare the effects of using saline and Plasma-Lyte for fluid therapy in patients undergoing major surgery in our teaching hospital. We found that that the cluster crossover design and opt-out patient consent process were logistically feasible and pragmatic with a 100% patient enrolment and 100% clinician acceptance rate. Significant contributing factors for the high level of patient enrolment included using an opt-out consent method, which provided both researchers and participants with a simple and convenient enrolment process. The logistical and operation success of the study was attributed to having a dedicated pharmacy-led governance team who were solely responsible for fluid storage and supply. The main study end point (i.e., renal function) as assessed by measurements of creatinine concentration was successfully achieved in 89.3% of participants. Further, given the pre-specified wash-in and wash-out periods, there were no crossover or contamination events; neither were there any safety incidents related to each specific fluid.

We found the volume of fluid infusion was low and that hyperchloraemia and acidosis were very mild and observed in the saline group. Compared with saline, Plasma-Lyte did not significantly affect the incidence of the primary outcome measure (postoperative AKI) within 72 hours, nor did it affect the incidence of other postoperative complications in that period. However, after adjustment, saline was associated with an increased duration of hospital stay (approximately one extra day). In alignment with prior literature [5–11], the administration of

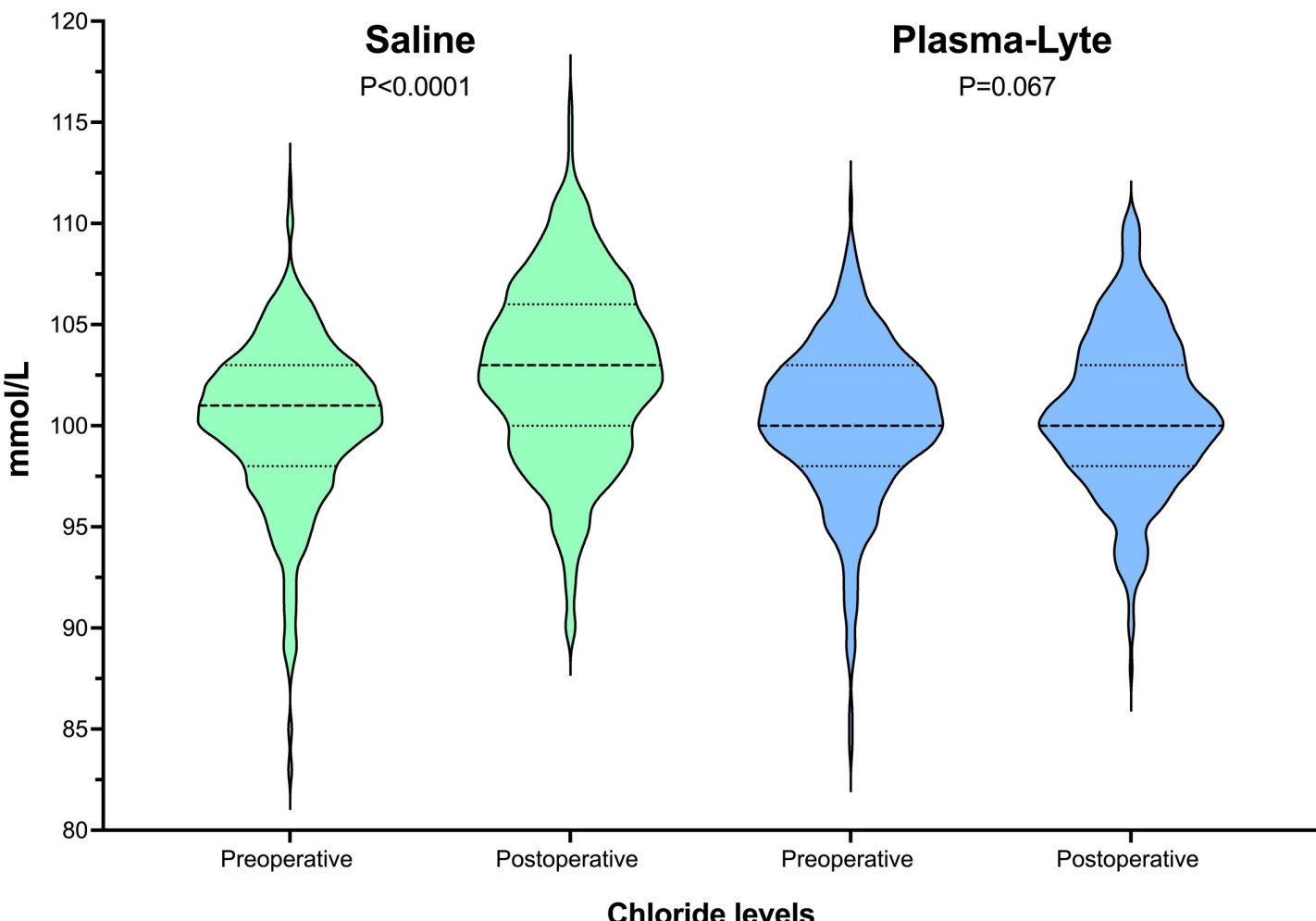

**Fig 3. Preoperative and postoperative changes in chloride levels.** The single thicker dashed lines represents the median value, and the thinner dashed lines represent the interquartile ranges.

saline was associated with higher chloride and lower bicarbonate levels (hyperchloraemic metabolic acidosis).

Our findings imply that crossover design studies investigating fluid therapy in the perioperative setting are feasible. Moreover, our findings suggest that clinicians can reasonably use either solution intraoperatively for adult patients undergoing major emergency or elective procedures if the volume of fluid administered is less than 2000 mL. However, concerns about the potentially harmful effects of hyperchloraemia remain. The clinical practice of using IV fluid solutions with unphysiological concentrations of chloride has been strongly discouraged due to the significant association between hyperchloraemia and the development of metabolic acidosis and adverse renal outcomes [30–32]. Moreover, for clinical trials investigating the effects of hyperchloraemic versus balanced solutions, clinical equipoise has also been recently questioned for the continued use of hyperchloraemic solutions in patients who develop hyperchloraemia [30]. Given that numerous studies have shown a harm signal associated with hyperchloraemic crystalloid solutions, without a comparable benefit signal [33], the ethical integrity of clinical trials comparing saline to balanced solutions has also come under scrutiny [30].

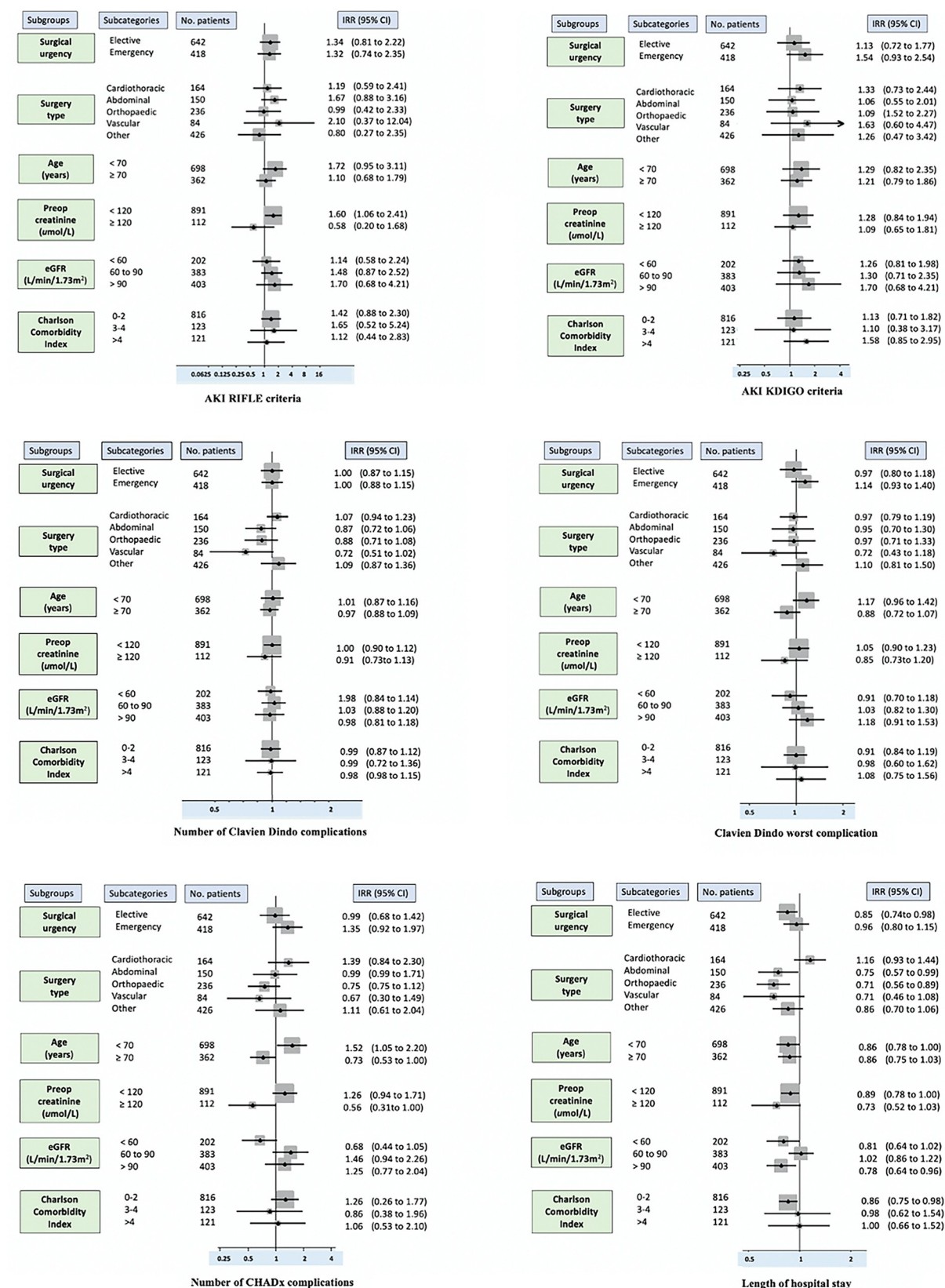

**Fig 4. Incidence rate ratios of key outcomes for patients receiving Plasma-Lyte compared to Saline.**

In the present study, we acknowledge that the low chloride load associated with the administration of small volumes of saline was well tolerated by our patient cohort, because they were at low risk for AKI. Moreover, we also acknowledge that our findings are not generalisable to patients with pre-existing renal dysfunction, those with metabolic acidosis and those receiving large volumes of hyperchloraemic solutions [31]. Recent editorials have emphasised that renal function is frequently unknown during surgery, and the volume of fluid replacement is unpredictable in acute care settings [31]. Therefore, because there is no evidence of benefit from hyperchloraemic solutions (other than in the management of metabolic alkalosis or hyponatraemia) and there is potential harm, there appears to be no physiologic or clinical justification for the continued use of "unphysiological" saline (or any other unbalanced crystalloid solution) for perioperative volume therapy, especially when safer and more physiological solutions are available [31–33].

## Relationship to previous studies

To the best of our knowledge, this is the first prospective, double-blinded study to compare saline and Plasma-Lyte for fluid therapy in major surgery patients. Previous observational studies have associated the use of saline with an increased risk of AKI in surgical patients [5–11]. Changes in serum chloride concentration, independent of serum sodium and bicarbonate, are associated with increased risk of AKI. Hyperchloraemia-associated AKI is thought to be related to renal vasoconstriction mediated by tubulo-glomerular feedback, and possibly other mechanisms, given the emerging role of chloride in regulating renal blood flow, glomerular filtration and tubular injury [34]. However, such studies outlines above are affected by selection bias and have not accounted for confounders, making causal inferences problematic. In the intensive care setting, two prospective fluid intervention pilot studies–the "Saline vs. Plasma-Lyte 148 for ICU fluid Therapy (SPLIT-ICU) and the "Saline against Lactated Ringer's or Plasma-Lyte" (SALT) trials [12, 13] evaluating the effects of buffered crystalloid solutions versus those of saline, reported no differences in the rate of major adverse kidney events between patient groups. Patients in the SALT trial [13] were admitted to the ICU mainly for sepsis or respiratory failure, while the SPLIT trial [12] largely studied surgical patients. Both studies, like ours, had low fluid administration volumes, with median volumes of less than 2000 mL administered in both arms. Notably, the subgroup in the SALT study who received larger volumes of fluid demonstrated a significant difference in the major adverse kidney events rates within 30 days rate between treatment arms. More recently, in contrast to our findings and those of the SALT and SPLIT trials, two large multiple crossover trials, one involving 13,347 non–critically ill [16] and another involving 15,802 critically ill [17] adult patients found that the use of balanced crystalloids (lactated Ringer's solution or Plasma-Lyte) resulted in a lower rate of renal injury compared to use of saline. However, these studies did not discriminate major surgery patients and were not double-blinded. In both studies, patients receiving saline demonstrated an increased incidence of hyperchloraemia and metabolic acidosis. Semler et al. [17] reported that postoperative surgical patients were less likely to develop major adverse kidney events than other subgroups, although such patients were not further characterised. Hospital length of stay was not studied in the intensive care study and was not a significant primary outcome in the emergency study. Pooled meta-analysis of studies performed in the critical care setting has not yet identified a difference in in-hospital mortality, incidence of AKI and the need for new renal replacement therapy between patients receiving balanced crystalloids and those receiving saline [35].

Regarding major surgery, the "Limiting I.V. Chloride to Reduce AKI" (LICRA) clinical trial examined the effects of restricting perioperative IV chloride use for kidney injuries [15]. In

this pragmatic, prospective study, data from 1139 cardiac patients revealed that a perioperative fluid strategy to restrict IV chloride administration was not associated with an altered incidence of AKI or other renal injury metrics. More recently, the "Saline or Lactated Ringer's" (SOLAR) trial [18] found no clinically meaningful difference in the risk of in-hospital mortality and major postoperative complications, including renal complications, among 8616 patients undergoing elective orthopaedic and colorectal surgery who received unblinded saline or lactated Ringer's solution. This trial reported AKI as a secondary outcome, and postoperative AKI classified as Acute Kidney Injury Network stage I–III occurred in 6.6% of lactated Ringer's patients versus 6.2% of saline patients. This incidence is lower than we observed, with an estimated relative risk of 1.18. The absolute differences between the treatment groups for AKI and all other outcomes were not clinically meaningful, at less than 0.5%.

## Strengths and limitations

This study involved a heterogenous group of patients to achieve wider applicability to the kind of patients likely to undergo the observed surgeries in a teaching hospital. The advantage of using an opt-out method together with a cluster crossover design provided both researchers and participants a simple and convenient enrolment process which yielded high enrolment rates. Moreover, this design may have resulted in less biased ascertainment of cases. We demonstrated the feasibility of conducting a hospital-wide study in the operating theatre and postoperative wards, including the ICU. Although this study recruited over 1000 patients, their baseline characteristics showed some imbalances. A higher proportion of patients in the saline group underwent emergency surgery and were male.

We acknowledge that AKI is assessed on the basis of creatinine concentrations and urine output, and we did not measure urine output. Therefore, our findings are more accurately reflective of changes in renal function, as assessed by measurements of creatinine concentrations. However, the observation that the AKI (defined only by changes in creatinine) point estimate favoured saline suggests that the lack of a Plasma-Lyte-mediated renal protective effect in this population is unlikely to represent a false negative finding. A relatively small average volume of fluid was administered and included a patient population with a relatively low risk of any stage of AKI. Renal function was tested at the discretion of the treating clinicians; accordingly, 113 patients (10.6%) did not undergo complete renal function tests. These patients were younger, had less comorbidities, had fewer postoperative complications and had a median hospital length of stay of one day, reflecting their low risk profile. It is reasonable to accept that these patients did not have a postoperative AKI. Moreover, a similar proportion of patients in each group (~90%) underwent complete renal function tests as part of their routine clinical care.

We recognise the inability of a single-cluster single-crossover design to address cluster and period effects. We did not analyse differences between the proportions of patients excluded from each cluster period, nor did we undertake a sensitivity analysis of the primary outcome among all patients who received the blinded study fluid i.e., not limited to those who met post-enrolment eligibility criteria. We also recognise the imbalance in the total number of patients in each study group, which is likely a product of the single crossover design. Finally, we did not assess blinding i.e., whether the clinicians using the fluid could correctly identify whether the allocated fluid was saline or Plasma-Lyte. We acknowledge these factors as limitations of the study.

Although this is a single-centre study, the study hospital has all the typical features of large tertiary hospitals in resource-rich countries. Therefore, the findings are likely to be relevant to similar institutions. Reassuringly, the incidence of AKI remains low and similar to that of our

prior study, which contained similar inclusion and exclusion criteria [11]. In that study, total fluid until AKI was higher in the AKI group. The amount of trial fluid in the present study was limited to a median value of 2000 mL. Had more fluid been administered, a difference between the effects of saline and Plasma-Lyte may have emerged. However, the use of this specific amount of fluid limits the relevance of our findings to this specific population. Finally, the observation that saline therapy was associated with an increased duration of hospital stay even after adjustment may be a chance finding, especially given the lack of differences in AKI or other complications. However, saline-treated patients exhibited a greater incidence of hyperchloraemia and acidosis, and such metabolic changes may have contributed to additional interventions and greater postoperative treatment duration. Finally, whilst the trial was completed in 2015, the collection and checking of data was undertaken manually, crosschecked by two independent persons, and then reaudited, a process which took almost two years to complete. Additional delays are attributed to the manuscript being rejected at other high impact journals and the coronavirus pandemic in 2020. We do not think this impacts on the integrity or interpretations of our findings.

## Conclusions

In a double-blind cluster crossover study comparing the renal effects of saline and Plasma-Lyte for fluid therapy in patients undergoing major elective and emergency surgery, we found that the study design was feasible to support a future follow-up larger clinical trial. Both patient and clinician acceptance were excellent. We observed no statistically significant difference in the incidence of postoperative AKI (as measured by changes in serum creatinine) or other postoperative complications. These findings can inform clinicians' decisions regarding which type of crystalloid fluid to use for patients receiving major surgery. Our findings also imply that clinicians can reasonably use either solution intraoperatively for adult patients undergoing major surgery.

## Supporting information

**S1 Checklist.**
(PDF)

**S1 Fig.**
(JPG)

**S1 Table. Physiochemical differences of the types of fluids administered compared to human plasma.**
(DOCX)

**S2 Table. Differences in perioperative variables in patients with measured and unmeasured renal function tests.**
(DOCX)

**S1 Database.**
(XLSX)

**S1 Protocol. Clinical trial protocol.**
(PDF)

## Author Contributions

**Conceptualization:** Laurence Weinberg, Rinaldo Bellomo.

**Data curation:** Laurence Weinberg, Michael Hua-Gen Li, Christopher Macgregor.

**Formal analysis:** Laurence Weinberg, Leonid Churilov.

**Funding acquisition:** Michael Hua-Gen Li, Rinaldo Bellomo.

**Investigation:** Laurence Weinberg, Michael Hua-Gen Li, Rinaldo Bellomo.

**Methodology:** Laurence Weinberg, Rinaldo Bellomo.

**Project administration:** Laurence Weinberg.

**Resources:** Christopher Macgregor, Kent Garrett, Jade Eyles.

**Supervision:** Laurence Weinberg, Leonid Churilov, Rinaldo Bellomo.

**Validation:** Laurence Weinberg.

**Visualization:** Laurence Weinberg, Leonid Churilov.

**Writing – original draft:** Laurence Weinberg, Michael Hua-Gen Li, Leonid Churilov, Christopher Macgregor, Rinaldo Bellomo.

**Writing – review & editing:** Laurence Weinberg, Michael Hua-Gen Li, Leonid Churilov, Christopher Macgregor, Kent Garrett, Jade Eyles, Rinaldo Bellomo.

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
