## [Decision Letter · Decision Letter 0]

4 Jan 2021

PONE-D-20-37578

The effects of 0.9% saline versus Plasma-Lyte 148 on acute kidney injury in patients undergoing major surgery: a single-centre double-blinded cluster crossover trial

PLOS ONE

Dear Dr. Weinberg,

Thank you for submitting your manuscript to PLOS ONE. After careful consideration, we feel that it has merit but does not fully meet PLOS ONE’s publication criteria as it currently stands. Therefore, we invite you to submit a revised version of the manuscript that addresses the points raised during the review process.

Some methodological, statistical and mechanistics limitations raised by reviewers should be adequately addressed 

We look forward to receiving your revised manuscript.

Kind regards,

Felipe Dal Pizzol

Academic Editor

PLOS ONE

Journal Requirements:

3.Thank you for stating the following in the Competing Interests section:

"I have read the journal's policy and the authors of this manuscript have the following competing interests: Prof Rinaldo Bellomo and A/Prof Laurence Weinberg have received honoraria of <US$5000 from Baxter Healthcare for consulting activities. The Australian and New Zealand Intensive Care Research Centre and the Departments of Intensive Care and Anaesthesia at Austin Health have received research grants from Baxter Healthcare. All aspects of the study design, execution, data collection, and analysis have been conducted independently of Baxter Health or any other industry."

Reviewers' comments:

Reviewer's Responses to Questions

**Comments to the Author**

1. Is the manuscript technically sound, and do the data support the conclusions?

Reviewer #1: Yes

Reviewer #2: Partly

Reviewer #3: Yes

Reviewer #4: Partly

2. Has the statistical analysis been performed appropriately and rigorously? 

Reviewer #1: Yes

Reviewer #2: I Don't Know

Reviewer #3: Yes

Reviewer #4: Yes

3. Have the authors made all data underlying the findings in their manuscript fully available?

Reviewer #1: Yes

Reviewer #2: No

Reviewer #3: Yes

Reviewer #4: Yes

4. Is the manuscript presented in an intelligible fashion and written in standard English?

Reviewer #1: Yes

Reviewer #2: Yes

Reviewer #3: Yes

Reviewer #4: Yes

5. Review Comments to the Author

Reviewer #1: Thank you for the opportunity to review your work.

This is one of the very fortunate cases where the reviewer job is easy. This is a solid pilot trial, well conducted, clearly reported and that is of crucial importance for larger endeavours.

Data reporting is clear. The observed imbalances are accounted for in the methods and discussion. The primary endpoint is properly analysed.

My only (very very minor) comment is that I would not be that sure if chloride is a possible culprit for AKI, so perhaps just lower the tone a bit by saying that balanced fluids may be associated with less AKI and that chloride could be related to it. I would specially avoid making statements about chloride and AKI in the abstract.

Reviewer #2: Important note: This review pertains only to ‘statistical aspects’ of the study and so ‘clinical aspects’ [like medical importance, relevance of the study, ‘clinical significance and implication(s)’ of the whole study, etc.] are to be evaluated [should be assessed] separately/independently. Further please note that any ‘statistical review’ is generally done under the assumption that (such) study specific methodological [as well as execution] issues are perfectly taken care of by the investigator(s). This review is not an exception to that and so does not cover clinical aspects {however, seldom comments are made only if those issues are intimately / scientifically related & intermingle with ‘statistical aspects’ of the study}. Agreed that ‘statistical methods’ are used as just tools here, however, they are vital part of methodology [and so should be given due importance].

COMMENTS: Actually, your ABSTRACT is well drafted but assay type [though you started with ‘Background’, I guess with intention to divide the ABSTRACT in sections]. Please note that it is preferable [refer to item 1b of CONSORT checklist 2010: Structured summary of trial design, methods, results, and conclusions] to divide the ABSTRACT with small sections like ‘Objective(s)’, ‘Methods’, ‘Results’, ‘Conclusions’, etc. which is an accepted practice of most of the good/standard journals [including this one]. It will definitely be more informative then, I think, whatever the article type may be.

I very strongly feel (and so have a very serious objection) that inclusion of the term ‘crossover trial’ is by [I guess] oversight/mistake. Note that the cross-over design {as per my knowledge of ‘standard definition’ and ‘ideal/original concept’} is a special case of a randomized control trial that allows each subject to serve as his own control. Following is a quote from a standard text-book:

In order to use the cross–over design, assumption to be made is that the effect of the intervention during the first period must not carry–over into the second period (carry–over effect or spill-over effect). Therefore, it is necessary that the therapies under study have no carry–over effect. Differential carry-over effect may be eliminated by interposing a long dry–out or wash–out period (the length of which is determined by the pharmacological properties of the drugs being tested) between the termination of the treatment given first and the beginning of the treatment given second.

I am sure that the authors know above quoted things. However, please specify clearly if you intended to imply something else (some other concept) by the term ‘crossover trial’ [remember that this is a scientific/academic document and so all details should be clearly/correctly communicated]. In fact, use of (standard) crossover trial in the field of ‘surgery’ is rare and so is highly appreciated [if it is correctly adapted/used]. Further read the following paragraph from the same book:

The appeal of this design is to avoid ‘between – subject’ variation in estimating the intervention effect, thereby all patients can be assured that sometime during the course of investigation, they will receive the new therapy {both treatments}. However, this method of study is not suitable if the drug of interest cures the disease, if the drug is effective only during a certain stage of the disease or if the disease changes radically during the period of time required for the study.

In table-2, you compare two treatment [Plasma-Lyte vs Saline after surgery] with different sample (group) sizes [in real crossover trial the group sizes are same as each subject will receive both treatments ultimately (only in different ‘order’)]. One known variation is ‘case-crossover’ design. This design is useful when the risk factor/exposure is transient. Data yielded by this design are analyzed as of ‘matched case-control’ studies. I do not see any of these [concept used as per standard] anywhere in this manuscript.

All the methodologically important concepts [like double-blinding, clustering, crossover trial] adapted/used here are not made clear. Account given {example regarding double-blinding described in lines 104-118 & also figure-1 regarding ‘Cluster crossover design of Saline vs Plasma-Lyte study’ does not depict clearly the ‘crossover’ nature, it mainly gives ‘time line’} is ‘confusing’, I feel. Advantages generally known of double-blinding are applicable for “parallel independent groups” design. For ‘crossover’ design (except avoidance of ‘auto- and hetero-suggestion’ bias) they may differ, I guess.

Moreover, all ‘statistical analyses’ performed here [example, line 164-65: Comparisons between categorical variables were made using chi-square and Fisher’s exact tests, while continuous variables were compared using the Mann–Whitney U test. including estimation of IRR, HR, all 95%CIs] assume ‘independence of the groups/samples’ which is violated in case ‘crossover trial’.

In my opinion, it [the ethics committee granting waiver of participant consent] is not a standard (practice) for any reason [like given in lines 77-79: participation in the research posed no more than minimal increase in risk to individuals than what they would be exposed to if they were not in the study, the ethics committee granted waiver of participant consent]. Other analyses {as described in lines 172-180 (We conducted time-to-event analyses using Cox proportional hazard regression models, and compared non-dichotomous outcomes using Poisson or negative binomial regression models, all adjusted for the same set of covariates, with treatment effects reported as adjusted hazard ratios with 95% CIs. The threshold for statistical significance was a P value of 0.05. All secondary and exploratory endpoints are reported as point estimates of treatment effects with 95% CIs.29 We report both covariate-adjusted and unadjusted outcomes. We used forest plots to present the primary and key secondary outcomes regarding the consistency of a treatment effect across the subgroups)} are alright.

Fig 2. CONSORT diagram and Fig 3. Incidence rate ratios of key outcomes for patients receiving Plasma-Lyte compared to Saline are also good. Overall, this article needs to be re-drafted and also change in title considering above highlighted points, I guess.

Reviewer #3: General Comments: In a single-cluster, single-crossover clinical trial, the authors compare administration of Plasma-Lyte versus 0.9% sodium chloride (saline) among adults undergoing surgery lasting more than two hours and requiring post-operative hospitalization for at least one night with regard to the primary outcome of AKI by RIFLE creatinine criteria. They report similar incidence of AKI (8.1% vs 10.7%) and complications but slightly shorter hospital length of stay with Plasma-Lyte compared to saline. Strengths include: (1) importance of the research question; (2) control of study group assignment by the trial; (3) control of fluid in the operating room, ICUs, and hospital wards. Weaknesses include: (1) the inability of a single-cluster single-crossover design to address cluster and period effects; (2) a population with a relatively low risk of any stage of AKI; (3) a relatively small average volume of fluid; and (3) presentation of the manuscript in a manner that de-emphasizes the pilot endpoints while over-emphasizing the clinical endpoints and incomplete description of the study population.

Major Comments:

1. Study population - The description of the methods in the text suggest that the blinded fluid assigned to a given period was applied to all patients in the OR, ICUs, and wards during the study period...and then only those who met the study's inclusion criteria were analyzed (i.e., patients could have received blinded study fluid and not be in the analysis). Is this correct? If so, this is reasonable for a comparison of the effectiveness of two commonly used interventions, but the methods and CONSORT diagram should make this clear. Specifically, did the 4,586 patients excluded from the primary analytic population because they had a surgical duration < 2 hours receive the blinded study fluid assigned to that cluster? If so, it is inaccurate to describe them as excluded from the trial prior to study group assignment. These patients should be excluded from each respective group AFTER allocation. This is important, in part, because this trial appears to have a primary analytic population that was defined using post-enrollment factors (e.g. duration of surgery -- something that cannot be determined prior to enrollment). This is usually a major problem in randomized trials, as study group assignment might then impact which patients meet the eligibility criteria, re-introducing selection bias (which randomization usually mitigates). To convince the readers that use of post-enrollment eligibility criteria did not bias the analysis, please consider: (1) moving all post-enrollment exclusions below allocation in the consort and confirming that the proportion of patients excluded for each criteria after group assignment did not differ between the two study groups; (2) adding (if available) a sensitivity analysis of the primary outcome among ALL patients who received the blinded study fluid -- not limited to those who met post-enrollment eligibility criteria. (If I have misunderstood the application of these eligibility criteria and they were truly assessed prior to enrollment and patients with shorter surgical durations or patients in the ICU who had not undergone a qualifying operation did NOT receive study fluid, please ignore this comment, but clarify in the methods).

2. Outcomes - Under Methods the Primary Aim of this study was described as "to establish the pilot feasibility, safety, and preliminary efficacy evidence base for a large trial". In the results section and conclusions, however, the authors focus on the clinical endpoints as if the primary aim of the trial were to guide clinical care (like the large trial they hope this pilot will support). For example, the first paragraph of the discussion's "key finding" is "our findings suggest that clinicians can reasonably use either solution intraoperatively...". In the conclusion they state "these findings can inform clinician's decision regarding which type of crystalloid fluid to use for patients receiving major surgery." A pilot trial aimed at establishing feasibility and safety of a larger trial (with a single-cluster single-crossover design that does not balance baseline characteristics or address cluster and period effects) should NOT be used to inform clinicians decision making in the treatment of patients! This is an excellent pilot study, but the authors must revise the results and discussion/conclusion to emphasize the feasibility, safety, and preliminary efficacy aims that they set out to address. What measures suggest this design is feasible? Enrollment rate? Compliance? Crossover or contamination? Rates of the outcome overall? Rare occurrence of the exclusion criteria? Clinician acceptence? Patient acceptence? What information informs the safety of a larger trial? Please present any information relevant to these assesments in the results. Please revise the discussion and conclusion to give the author's assessment, as the one's who know this trial best, on how this trial does inform the design of future trials on this topic in the field. How should a larger trial be designed similarly or differently? Population? Outcomes? Duration of intervention? Type of surgery? Please also make sure these changes to focus on the feasibility and safety aims of this pilot trial are reflected in the abstract.

3. Patient population - please add to the limitations section the imbalance in the total number of patients in each study group, which is likely a product of the single crossover (but readers must be reassured this is not a product of post-enrollment exclusions preferentially in one group which would imbalance the study populations)

4. Please clarify that patients undergoing a second surgery did not receive the study fluid during that surgery but they were still in the intention-to-treat population from their initial surgery, right? You did not throw out their initial enrollment, correct?

5. Please add a statement to the discussion regarding why this clinical trial completed in February of 2015 is being made publicly available more than 5 years later. This happens in research, but funding agencies and other authorities are increasingly mandate release of RCT results within 12 months of completion -- so some note of what difficulties were encountered that precluded this will help readers understand the context.

6. Did you assess the success of the blinding? If yes, please present results. If not, please acknolwedge in discussion as a limitation. (As you know, saline and balanced crystalloids have over effects on commonly measured labs and blinding is especailly difficult to maintain in a cluster-level trial where all patients in the block are on the same fluid assignment)

7. Outcome assessment - please present more clearly in the results the number and proportion of patients in each group who did not have a creatinine measured with which to assess post-operative AKI (primary outcome) -- and a P value to show that this proportion was the same between groups. This shows up in the discussion but I don't see it in the results.

8. Please describe the process by which complications were graded by two indpendent clinicians

9. In the discussion, "7942" should be "15802" when referring to the SMART trila

Reviewer #4: The authors compared the effects of saline to those of Plasmalyte in patients after cardiac surgery. In this double-blinded cluster, crossover trial, they enrolled 602 patients to the saline group and 458 to the Plasmalyte group. There was no major difference in biochemical findings.

The study is somewhat old, but this should not influence the findings.

Major comments

1. The authors seem to ignore the basic mechanism by which saline solutions may alter the renal function, i.e. the hyperchloremia, with the secondary metabolic acidosis. Saline-treated patients had peak serum chloride levels at 103 [100 to 106] mmol/l vs 101 [98 to 103] mmol/L), there is no reason to believe in any difference in renal function. The conclusions of the present study are that the administration of 2 L of either saline or Plasmalyte does not affect the chloride levels and therefore cannot result in any difference in renal function, as assessed by the creatinine concentration. That is interesting.

The figure 2 is useless, and could go in the supplement material. A figure showing the time course of chloride and bicarbonate is the essence of the paper.

2. Since AKI is assessed on the basis of creatinine concentrations and urine output, and the authors did not measure urine output, this is not AKI (I know some other papers did that, but this is not a reason to repeat the same mistakes). The authors should refer to a study of the renal function, as assessed by measurements of creatinine concentrations.

Minor comment

The discussion section should briefly explain how the saline-induced hyperchloremia can influence renal function.

6. PLOS authors have the option to publish the peer review history of their article (what does this mean?). If published, this will include your full peer review and any attached files.

Reviewer #1: **Yes: **Fernando G Zampieri

Reviewer #2: No

Reviewer #3: No

Reviewer #4: **Yes: **JL Vincent

---

## [Author Response · Author response to Decision Letter 0]

22 Feb 2021

RESPONSE TO REVIEWERS

PONE-D-20-37578

The effects of 0.9% saline versus Plasma-Lyte 148 on renal function as assessed by creatinine concentrations in patients undergoing major surgery: a single-centre double-blinded cluster crossover trial

On behalf of my co-authors, I would like to sincerely thank each Reviewer for considering our revised manuscript for publication. The comprehensive reviews and constructive comments provided by yourself and the expert reviewers have been immensely appreciated. The insightful and constructive comments have enabled us to strengthen the scientific rigor and merits of our research. Thank you once again for considering our resubmission for publication. 

Reviewer #1

We thank Professor Fernando Zampieri for his valuable and constructive comments. Further we appreciate the time taken to review our manuscript

Reviewer #1, Question 1. This is one of the very fortunate cases where the reviewer job is easy. This is a solid pilot trial, well conducted, clearly reported and that is of crucial importance for larger endeavours. Data reporting is clear. The observed imbalances are accounted for in the methods and discussion. The primary endpoint is properly analysed.

Question 1. Authors’ response: Thank you for your positive comments. Once again, we thank you taking the time to review our manuscript. 

Reviewer #1, Question 2. My only (very very minor) comment is that I would not be that sure if chloride is a possible culprit for AKI, so perhaps just lower the tone a bit by saying that balanced fluids may be associated with less AKI and that chloride could be related to it. I would specially avoid making statements about chloride and AKI in the abstract.

Question 2. Authors’ response: Thank you for these important comments. In the abstract, we have removed all references about any association with chloride and AKI. 

In the “Objectives” section we now state “In a double-blind cluster crossover study comparing the renal effects of saline and Plasma-Lyte for fluid therapy in patients undergoing major elective and emergency surgery, we found that the study design was feasible to support a future follow-up larger clinical trial. Both patient and clinician acceptance were excellent. We observed no statistically significant difference in the incidence of postoperative AKI (as measured by changes in serum creatinine) or other postoperative complications.”

Reviewer #2

We thank Reviewer #2 for their valuable time taken to review our manuscript. Please find below a detailed response to each of the important comments. 

Reviewer #2: Important note: This review pertains only to ‘statistical aspects’ of the study and so ‘clinical aspects’ [like medical importance, relevance of the study, ‘clinical significance and implication(s)’ of the whole study, etc.] are to be evaluated [should be assessed] separately/independently. 

Reviewer #2: Question 1. COMMENTS: Actually, your ABSTRACT is well drafted but assay type [though you started with ‘Background’, I guess with intention to divide the ABSTRACT in sections]. Please note that it is preferable [refer to item 1b of CONSORT checklist 2010: Structured summary of trial design, methods, results, and conclusions] to divide the ABSTRACT with small sections like ‘Objective(s)’, ‘Methods’, ‘Results’, ‘Conclusions’, etc. which is an accepted practice of most of the good/standard journals [including this one]. It will definitely be more informative then, I think, whatever the article type may be.

Authors’ response, Question 1. Thank you for this important comment. We have now followed the CONSORT checklist and restructured the Abstract into the following 4 headings: Objectives, Methods, Results, and Conclusions.

Reviewer #2: Question 2. I very strongly feel (and so have a very serious objection) that inclusion of the term ‘crossover trial’ is by [I guess] oversight/mistake. Note that the cross-over design {as per my knowledge of ‘standard definition’ and ‘ideal/original concept’} is a special case of a randomized control trial that allows each subject to serve as his own control. Following is a quote from a standard text-book: In order to use the cross–over design, assumption to be made is that the effect of the intervention during the first period must not carry–over into the second period (carry–over effect or spill-over effect). Therefore, it is necessary that the therapies under study have no carry–over effect. Differential carry-over effect may be eliminated by interposing a long dry–out or wash–out period (the length of which is determined by the pharmacological properties of the drugs being tested) between the termination of the treatment given first and the beginning of the treatment given second.

I am sure that the authors know above quoted things. However, please specify clearly if you intended to imply something else (some other concept) by the term ‘crossover trial’ [remember that this is a scientific/academic document and so all details should be clearly/correctly communicated]. In fact, use of (standard) crossover trial in the field of ‘surgery’ is rare and so is highly appreciated [if it is correctly adapted/used]. Further read the following paragraph from the same book: The appeal of this design is to avoid ‘between – subject’ variation in estimating the intervention effect, thereby all patients can be assured that sometime during the course of investigation, they will receive the new therapy {both treatments}. However, this method of study is not suitable if the drug of interest cures the disease, if the drug is effective only during a certain stage of the disease or if the disease changes radically during the period of time required for the study.

Authors’ response, Question 2. That you for this excellent comment. We strongly agree with your comments about a “cross-over design” study. This is a very important point, however it’s important to appreciate that the design of our study is not a conventional “cross-over trial”, rather, as we have clearly stated in the title, abstract and the methods sections of the manuscript, our design is a “cluster crossover trial”. 

There are fundamental and important differences in the analysis of a “cross-over design study” and a “cluster crossover trial” (1). In our study the whole “cluster” of individuals (i.e. the hospital), rather than the individual patients, has been crossed over from treatment A to treatment B with the order of treatments being randomly chosen. All patients admitted to our single institution over the predefined 6-week period can therefore be thought of as belonging to a cluster. The cluster cross-over nature of our study would have perhaps been clearer if multiple clusters/centres were to be crossed-over such as in, e.g., a stepped-wedge design, but the fact that in our study only a single cluster underwent a cross-over does not discount the single cluster single cross-over nature of the study.

The statistical analysis applied for our cluster crossover trial are therefore appropriate. 

Reference

1Arnup SJ, McKenzie JE, Hemming K, Pilcher D, Forbes AB. Understanding the cluster randomised crossover design: a graphical illustration of the components of variation and a sample size tutorial. Trials. 2017 Aug 15;18(1):381. Doi: 10.1186/s13063-017-2113-2. PMID: 28810895; PMCID: PMC5557529.

Reviewer #2: Question 3. In table-2, you compare two treatment [Plasma-Lyte vs Saline after surgery] with different sample (group) sizes [in real crossover trial the group sizes are same as each subject will receive both treatments ultimately (only in different ‘order’)]. One known variation is ‘case-crossover’ design. This design is useful when the risk factor/exposure is transient. Data yielded by this design are analyzed as of ‘matched case-control’ studies. I do not see any of these [concept used as per standard] anywhere in this manuscript.

Authors’ response, Question 3. As we have stated above, given that we have performed a “cluster crossover trial”, we believe our statistical analyses are appropriate. 

Reviewer #2, Question 4. All the methodologically important concepts [like double-blinding, clustering, crossover trial] adapted/used here are not made clear. Account given {example regarding double-blinding described in lines 104-118 & also figure-1 regarding ‘Cluster crossover design of Saline vs Plasma-Lyte study’ does not depict clearly the ‘crossover’ nature, it mainly gives ‘time line’} is ‘confusing’, I feel. Advantages generally known of double-blinding are applicable for “parallel independent groups” design. For ‘crossover’ design (except avoidance of ‘auto- and hetero-suggestion’ bias) they may differ, I guess.

Authors’ response, Question 4. Thank you for this excellent comment. We have now included an additional statement in the methods section clarifying what a “cluster crossover trial” is. We agree that this will be important for the reader who is not familiar with this type of specific trial design. Further, we have made minor modifications to Figure 1, to allow the reader to better appreciate the cluster crossover design. In the Methods section, we also have included the following an additional statement “Unlike a conventional randomized clinical trial, “clusters” of individuals, rather than the individual patients were blindly allocated to receive either saline or Plasma-Lyte”. 

We have also made some minor modification to Figure 1 to provide the reader with more clarity regarding the cluster crossover trial design.

Reviewer #2, Question 5. Moreover, all ‘statistical analyses’ performed here [example, line 164-65: Comparisons between categorical variables were made using chi-square and Fisher’s exact tests, while continuous variables were compared using the Mann–Whitney U test. Including estimation of IRR, HR, all 95%Cis] assume ‘independence of the groups/samples’ which is violated in case ‘crossover trial’.

Authors’ response, Question 5. As we have stated above, given that we have performed a “cluster crossover trial” with a single cluster and not an individual patient “cross-over trial”, we believe our statistical analysis are appropriate. 

Reviewer #2, Question 6. In my opinion, it [the ethics committee granting waiver of participant consent] is not a standard (practice) for any reason [like given in lines 77-79: participation in the research posed no more than minimal increase in risk to individuals than what they would be exposed to if they were not in the study, the ethics committee granted waiver of participant consent]. Other analyses {as described in lines 172-180 (We conducted time-to-event analyses using Cox proportional hazard regression models, and compared non-dichotomous outcomes using Poisson or negative binomial regression models, all adjusted for the same set of covariates, with treatment effects reported as adjusted hazard ratios with 95% Cis. The threshold for statistical significance was a P value of 0.05. All secondary and exploratory endpoints are reported as point estimates of treatment effects with 95% Cis.29 We report both covariate-adjusted and unadjusted outcomes. We used forest plots to present the primary and key secondary outcomes regarding the consistency of a treatment effect across the subgroups)} are alright.

Authors’ response, Question 6. Thank you for the positive comments regarding our use of these statistical tests. 

Reviewer #2, Question 7. Fig 2. CONSORT diagram and Fig 3. Incidence rate ratios of key outcomes for patients receiving Plasma-Lyte compared to Saline are also good. 

Authors’ response, Question 7. Thank you for these positive comments. 

Reviewer #2, Question 8. Overall, this article needs to be re-drafted and also change in title considering above highlighted points, I guess.

Authors’ response, Question 8. We hope our responses have assured the Reviewer that our study is not a “cross-over trial”, rather a “cluster crossover trial”. Therefore, we believe our statistical analyses are appropriate. 

Reviewer #3

We thank Reviewer #3 for their valuable comments and time taken to review our manuscript. Further we appreciate this expert and considered review. Please find below a detailed response to each of these important comments. 

Reviewer #3: Question 1. General Comments: In a single-cluster, single-crossover clinical trial, the authors compare administration of Plasma-Lyte versus 0.9% sodium chloride (saline) among adults undergoing surgery lasting more than two hours and requiring post-operative hospitalization for at least one night with regard to the primary outcome of AKI by RIFLE creatinine criteria. They report similar incidence of AKI (8.1% vs 10.7%) and complications but slightly shorter hospital length of stay with Plasma-Lyte compared to saline. 

Strengths include: 

(1) importance of the research question; 

(2) control of study group assignment by the trial; 

(3) control of fluid in the operating room, ICUs, and hospital wards. 

Weaknesses include: 

(1) the inability of a single-cluster single-crossover design to address cluster and period effects; 

(2) a population with a relatively low risk of any stage of AKI; 

(3) a relatively small average volume of fluid; and 

(3) presentation of the manuscript in a manner that de-emphasizes the pilot endpoints while over-emphasizing the clinical endpoints and incomplete description of the study population.

Authors’ response, Question 1. Thank you for highlighting the key strengths and weaknesses of the manuscript. We hope that our detailed responses and modification to these insightful questions enhance the scientific merits of this study and address and strengthen some of the weaknesses highlighted above. 

Reviewer #3: Major Comments:

Reviewer #3: Question 1. Study population – The description of the methods in the text suggest that the blinded fluid assigned to a given period was applied to all patients in the OR, ICUs, and wards during the study period...and then only those who met the study’s inclusion criteria were analyzed (i.e., patients could have received blinded study fluid and not be in the analysis). Is this correct? If so, this is reasonable for a comparison of the effectiveness of two commonly used interventions, but the methods and CONSORT diagram should make this clear. Specifically, did the 4,586 patients excluded from the primary analytic population because they had a surgical duration < 2 hours receive the blinded study fluid assigned to that cluster? If so, it is inaccurate to describe them as excluded from the trial prior to study group assignment. These patients should be excluded from each respective group AFTER allocation. This is important, in part, because this trial appears to have a primary analytic population that was defined using post-enrolment factors (e.g., duration of surgery – something that cannot be determined prior to enrolment). This is usually a major problem in randomized trials, as study group assignment might then impact which patients meet the eligibility criteria, re-introducing selection bias (which randomization usually mitigates). 

Authors’ response, Question 1. The reviewer has made some excellent points, all of which we agree with. Importantly, as the Reviewer has stated, for pragmatic and logistical reasons, blinded fluid was assigned for a given period and administered to all patients undergoing surgery during that period. We then “included” or “excluded” patients based on prespecified trial criteria, which have been outlined in our manuscript. The fluid was then continued in all patients in the intensive care unit and all patients on the ward. Importantly, we only analysed patients who met the study’s inclusion criteria.

We also agree with Reviewer that this trial has a primary analytic population defined using post-enrolment factors (e.g., duration of surgery), which as the Reviewer has correctly stated, could not be determined prior to enrolment. We also agree that this can be a concern in randomized trials, as study group assignment can potentially impact on which patients meet the eligibility criteria, re-introducing selection bias (which randomization usually mitigates). However, for our study, we acknowledge that this is “theoretically” possible i.e., that the intervention could bias the inclusion and exclusion criteria, however in reality, there is no causality whatsoever between the treatment and the inclusion/exclusion criteria, all of which we prespecified. We assure the reviewer that the treatment i.e., saline or Plasmalyte does bias patient inclusion or patient exclusion. 

In response to the Reviewer’s comments, in the Results section, we now state “In total, 5646 patients were screened over the duration of the two predefined cluster periods. During the first 6-week cluster period, 2933 patients were allocated to receive saline; during the second 6-week cluster period, 2713 patients were allocated to receive Plasma-Lyte. After exclusions, 1060 patients fulfilled the inclusion criteria – 602 patients were allocated to the saline group and 458 to the Plasma-Lyte group”.

In addition, our CONSORT diagram has now been corrected to accurately reflect this important point. Thank you once again for these excellent suggestions. 

Reviewer #3: Question 2. To convince the readers that use of post-enrolment eligibility criteria did not bias the analysis, please consider: Moving all post-enrolment exclusions below allocation in the consort and confirming that the proportion of patients excluded for each criteria after group assignment did not differ between the two study groups

Authors’ response, Question 2. We have now moved all post-enrolment exclusions below allocation in the Consort diagram. Kindly refer to the revised Consort diagram presented above. We did not analyse differences between the proportions of patients excluded from each cluster period. This was not the proposed intention of the study and we did not have sufficient funding nor the resources to undertake this. We do not think this compromises the integrity of our findings. We have now acknowledged this as a limitation of the study in the discussion section. 

Reviewer #3: Question 3. Adding (if available) a sensitivity analysis of the primary outcome among ALL patients who received the blinded study fluid – not limited to those who met post-enrolment eligibility criteria. (If I have misunderstood the application of these eligibility criteria and they were truly assessed prior to enrolment and patients with shorter surgical durations or patients in the ICU who had not undergone a qualifying operation did NOT receive study fluid, please ignore this comment, but clarify in the methods).

Authors’ response, Question 3. We did not analyse differences between the proportions of patients excluded from each cluster period. Neither did we undertake a sensitivity analysis of the primary outcome among all patients who received the blinded study fluid i.e., not limited to those who met post-enrolment eligibility criteria. We have acknowledged these limitations of the study and outlined these in the discussion section.

Reviewer #3: Question 4. Outcomes – Under Methods the Primary Aim of this study was described as “to establish the pilot feasibility, safety, and preliminary efficacy evidence base for a large trial”. In the results section and conclusions, however, the authors focus on the clinical endpoints as if the primary aim of the trial were to guide clinical care (like the large trial they hope this pilot will support). For example, the first paragraph of the discussion’s “key finding” is “our findings suggest that clinicians can reasonably use either solution intraoperatively...”. In the conclusion they state “these findings can inform clinician’s decision regarding which type of crystalloid fluid to use for patients receiving major surgery.” 

A pilot trial aimed at establishing feasibility and safety of a larger trial (with a single-cluster single-crossover design that does not balance baseline characteristics or address cluster and period effects) should NOT be used to inform clinicians decision making in the treatment of patients! This is an excellent pilot study, but the authors must revise the results and discussion/conclusion to emphasize the feasibility, safety, and preliminary efficacy aims that they set out to address. 

Patient acceptance? What information informs the safety of a larger trial? What measures suggest this design is feasible? Enrolment rate? Compliance? Crossover or contamination? Rates of the outcome overall? Rare occurrence of the exclusion criteria? 

Clinician acceptance? Patient acceptance? What information informs the safety of a larger trial? 

Please present any information relevant to these assessments in the results. 

Authors’ response, Question 4. Thank you for these excellent points that we agree with. Accordingly, we have revised the results, discussion and conclusion section of the study and included detailed information on study design feasibility, enrolment rate, compliance, crossover or contamination, rates of the outcome overall, occurrence of the exclusion criteria, and clinician and patient acceptance. 

We hope that we have addressed this point in detail. Below is a brief overview of the additional changes made. 

1. Abstract. We have included the following statement “The primary aim was to establish the pilot feasibility, safety and preliminary efficacy evidence base for a large interventional trial to establish whether saline or Plasma-Lyte is the preferred crystalloid fluid for managing major surgery patients. The primary efficacy outcome was the proportion of patients with changes in renal function as assessed by creatinine concentrations during their index hospital admission. We used changes in creatinine to define acute kidney injury (AKI) according to the RIFLE criteria.”

The results of the abstract also now address feasibility end points, efficacy end pints and safety end points. We now state “The study was feasible with 100% patient and clinician acceptance. There were no deviations from the trial protocol.” The results conclude with the statement “There were no serious adverse events that related to the specific type of fluid received and there were no fluid crossover or contamination events.” 

The conclusion has been modified reflect feasibility, efficacy and safety. We now state “The study design was feasible to support a future follow-up larger clinical trial. Patients treated with saline did not demonstrate an increased incidence of postoperative AKI (defined as changes in creatinine) compared to those treated with Plasma-Lyte. Our findings imply that clinicians can reasonably use either solution intraoperatively for adult patients undergoing major surgery.”

2. In the methods section we now briefly describe and overview the logistical and operation challenges, with reference to fluid governance, storage and delivery to all clinical areas. 

Further in the methods section we state “Feasibility outcomes included patient and clinician acceptance, recruitment rate, reasons for exclusion, logistical or operation feasibility with fluid storage and delivery to the clinical areas where the trials was undertaken, deviations from trial protocol, unblinding rate, and number of patients who fulfilled eligibility who did not have renal function measured as part of routine clinical care.” 

The primary efficacy outcome was the proportion of patients with AKI defined by creatinine levels and assessed according to the risk, injury, failure, loss of kidney function and end-stage renal failure (RIFLE) criteria during the index hospital admission.

We also state that “Safety outcomes included fluid administration errors, unintentional fluid discontinuation, crossover or contamination events.

3. Results. We systematically present the results of feasibility, efficacy and safety as outlined above. 

4. Conclusion section: As suggested we now address feasibility, preliminary efficacy and safety. We now conclude by stating “In a double-blind cluster crossover study comparing the renal effects of saline and Plasma-Lyte for fluid therapy in patients undergoing major elective and emergency surgery, we found that study design was feasible to support a future follow-up larger clinical trial. Both patient and clinician acceptance were excellent. We observed no statistically significant difference in the incidence of postoperative AKI (as measured by changes in serum creatinine and eGFR), or other postoperative complications.

Thank you once again for these excellent comments and insights. They are very much appreciated. 

Reviewer #3: Question 5. Please revise the discussion and conclusion to give the author’s assessment, as the one’s who know this trial best, on how this trial does inform the design of future trials on this topic in the field. How should a larger trial be designed similarly or differently? Population? Outcomes? Duration of intervention? Type of surgery? 

Authors’ response, Question 5. Thank you for this constructive comment. We have now made significant modifications to the discussion of the manuscript reflecting the above points. 

Reviewer #3: Question 6. Please also make sure these changes to focus on the feasibility and safety aims of this pilot trial are reflected in the abstract.

Authors’ response, Question 6. Thank you for this important comment. 

We have made the following changes to the Abstract. We include the following statement “The primary aim was to establish the pilot feasibility, safety and preliminary efficacy evidence base for a large interventional trial to establish whether saline or Plasma-Lyte is the preferred crystalloid fluid for managing major surgery patients. The primary efficacy outcome was the proportion of patients with changes in renal function as assessed by creatinine concentrations during their index hospital admission. We used changes in creatinine to define acute kidney injury (AKI) according to the RIFLE criteria.” 

The results of the abstract now address feasibility end points, efficacy end pints and safety end points. We now state “The study was feasible with 100% patient and clinician acceptance. There were no deviations from the trial protocol.” 

The results conclude with the statement “There were no serious adverse events that related to the specific type of fluid received and there were no fluid crossover or contamination events.” 

The conclusion has been modified reflect feasibility, efficacy and safety. We now state “The study design was feasible to support a future follow-up larger clinical trial. Patients treated with saline did not demonstrate an increased incidence of postoperative AKI (defined as changes in creatinine) compared to those treated with Plasma-Lyte. Our findings imply that clinicians can reasonably use either solution intraoperatively for adult patients undergoing major surgery.”

Reviewer #3: Question 7. Patient population – please add to the limitations section the imbalance in the total number of patients in each study group, which is likely a product of the single crossover (but readers must be reassured this is not a product of post-enrolment exclusions preferentially in one group which would imbalance the study populations)

Authors’ response, Question 7. Thank you for this excellent comment. In the limitations section we now state “We recognise the inability of a single-cluster single-crossover design to address cluster and period effects. We did not analyse differences between the proportions of patients excluded from each cluster period. Neither did we undertake a sensitivity analysis of the primary outcome among all patients who received the blinded study fluid i.e., not limited to those who met post-enrolment eligibility criteria. We also recognise the imbalance in the total number of patients in each study group, which is likely a product of the single crossover design. Finally, we did not assess blinding i.e., whether the clinicians using the fluid could correctly identify the whether the allocated fluid was saline or Plasma-Lyte. We acknowledge these factors as limitations of the study.”

Reviewer #3: Question 8. Please clarify that patients undergoing a second surgery did not receive the study fluid during that surgery but they were still in the intention-to-treat population from their initial surgery, right? You did not throw out their initial enrolment, correct?

Authors’ response, Question 8. This is an important comment, which we have addressed and clarifies in the methods section. We now state “Patients requiring a second operation during the trial period who initially received the study fluid, still received the study fluid in their reoperation. They were included in the intention-to-treat population from their initial surgery, however excluded from the study for their reoperation.”

Reviewer #3: Question 9. Please add a statement to the discussion regarding why this clinical trial completed in February of 2015 is being made publicly available more than 5 years later. This happens in research, but funding agencies and other authorities are increasingly mandate release of RCT results within 12 months of completion – so some note of what difficulties were encountered that precluded this will help readers understand the context.

Authors’ response, Question 9. Thank you for this question. Finally, whilst the trial was completed in 2015, the collection and checking of data was undertaken manually, crosschecked by two independent persons, and then reaudited, a process which took almost two years to complete. Additional delays can be attributed to the manuscript being rejected at other high impact journals and the coronavirus in 2020. Moreover, constructive review from reviewers at other journal also allowed us to address some of the limitation of the study and improve on the scientific merits of the study in its current submission to PLOS ONE. We do not think this delay impacts on the integrity or interpretation of our findings. We have briefly outlined this in the discussion section of the manuscript. 

Reviewer #3: Question 10. Did you assess the success of the blinding? If yes, please present results. If not, please acknowledge in discussion as a limitation. (As you know, saline and balanced crystalloids have over effects on commonly measured labs and blinding is especially difficult to maintain in a cluster-level trial where all patients in the block are on the same fluid assignment)

Authors’ response, Question 10. This is an excellent point that our research team were very conscious of for the duration of the study. Interestingly, we observed that clinicians using the trial fluid did not appear to make mention of what they thought the fluid was. In part, the cluster design resulted in consistency of the trial fluid being available at all times, and the fact that there was no available comparative crystalloid fluid to use was helpful. Importantly, when designing the labelling of the Trial fluids, we chose to name the Trial Fluid “Surgilyte” (see picture below, also included in the supplementary files), which also may have contributed to the high level of clinician acceptability. Initially, when designing this study, we thought it would be informative to formally survey clinicians using the fluid to assess how accurately they could identify the type of fluid used. We concluded that this may have had a negative impact with too much focus being projected on the type of fluid rather than maintain focus of the key study endpoints as we have outlined in the manuscript. 

In the limitations section we state “Finally, whilst the trial was completed in 2015, the collection and checking of data was undertaken manually, crosschecked by two independent persons, and then reaudited, a process which took almost two years to complete. Additional delays are attributed to the manuscript being rejected at other high impact journals and the coronavirus in 2020. We do not think this impacts on the integrity or interpretations of our findings.” 

Reviewer #3: Question 11. Outcome assessment – please present more clearly in the results the number and proportion of patients in each group who did not have a creatinine measured with which to assess post-operative AKI (primary outcome) – and a P value to show that this proportion was the same between groups. This shows up in the discussion but I don’t see it in the results.

Authors’ response, Question 11. Thank you for this important comment. We have added in the following statement to the results section: “Overall, there were 113 patients (10.6%) who were included in the trial who did not have their renal function measured as part of routine standard of care. The portion of patients in each group were similar (10.5% of patients receiving saline vs. 10.9% of patients receiving Plasma-Lyte, p = 0.841). 

Reviewer #3: Question 12. Please describe the process by which complications were graded by two independent clinicians

Authors’ response, Question 12. We have now updated this in the Methods section. We now state “Reporting and grading of complications were evaluated by two authors independently (MHL, CM) by undertaking an in-depth review of each patient’s clinical records. In the case of disagreement, the case was presented to two other authors (LW, RB) to reach consensus.”

Reviewer #3: Question 13. In the discussion, “7942” should be “15802” when referring to the SMART trial.

Authors’ response, Question 13. Thank you for picking up this important error. The SMART study enrolled 15,802 adults to receive saline or the balanced crystalloids (lactated Ringer’s solution or Plasma-Lyte A). The 7942 patients that was stated in our original submission is incorrect. This number was the number of patients who received the balanced-crystalloids group. This has now been corrected. 

Reviewer #4

We thank Professor JL Vincent for his valuable time taken to review our manuscript. Further, we appreciate his expert, constructive and considered review. Please find below a detailed response to each of the important comments. 

Reviewer #4, Question 1. The authors compared the effects of saline to those of Plasmalyte in patients after cardiac surgery. In this double-blinded cluster, crossover trial, they enrolled 602 patients to the saline group and 458 to the Plasmalyte group. There was no major difference in biochemical findings. The study is somewhat old, but this should not influence the findings.

Authors’ response, Question 1. Thank you for this question which has also been raised by Reviewer #3. Whilst the trial was completed in 2015, the collection and checking of data was undertaken manually, crosschecked by two independent persons, and then reaudited, a process which took almost two years to complete. Additional delays can be attributed to the manuscript being rejected at other high impact journals and the coronavirus in 2020. Moreover, constructive review from reviewers at other journal also allowed us to address some of the limitation of the study and improve on the scientific merits of the study in its current submission to PLOS ONE. We do not think this delay impacts on the integrity or interpretation of our findings. We have briefly outlined this in the discussion section of the manuscript. 

Major comments

Reviewer #4, Question 2. The authors seem to ignore the basic mechanism by which saline solutions may alter the renal function, i.e., the hyperchloremia, with the secondary metabolic acidosis. Saline-treated patients had peak serum chloride levels at 103 [100 to 106] mmol/l vs 101 [98 to 103] mmol/L), there is no reason to believe in any difference in renal function. The conclusions of the present study are that the administration of 2 L of either saline or Plasmalyte does not affect the chloride levels and therefore cannot result in any difference in renal function, as assessed by the creatinine concentration. That is interesting.

Authors’ response, Question 2. Thank you for this insightful question. In the introduction section of the manuscript, we state that controversy remains regarding the optimal fluid for use in this setting and that numerous studies in the past decade have examined the effects of hyperchloraemic solutions on renal function with most suggesting that balanced crystalloid solutions with lower chloride concentrations may have less nephrotoxic effects. We also now state in the introduction paragraph that “Hyperchloraemia has been reported to be associated with chloride-induced renal vasoconstriction, however, more recent studies, have reported that major morbidity, including AKI, is comparable between patients treated with saline and those treated with lactated balanced crystalloids. Other studies have yielded conflicting results.” These studies have also been referenced. 

Reviewer #4, Question 3. The figure 2 is useless and could go in the supplement material. A figure showing the time course of chloride and bicarbonate is the essence of the paper.

Authors’ response, Question 3. Given the comments made by Reviewer #3 (Question 1) pertaining to the Consort diagram, we respectfully think that the Consort diagram i.e., Figure 2 should remain in the manuscript. We have made significant modifications to Figure 2 that allows the reader to better understand the cluster crossover design used. 

Reviewer #4, Question 4. Since AKI is assessed on the basis of creatinine concentrations and urine output, and the authors did not measure urine output, this is not AKI (I know some other papers did that, but this is not a reason to repeat the same mistakes). The authors should refer to a study of the renal function, as assessed by measurements of creatinine concentrations.

Authors’ response, Question 4. This is an excellent comment, and we are very grateful for this suggestion. Accordingly, we have made numerous changes throughout the manuscript to reflect this change. This includes changes to the title, abstract, methods section, and discussion. 

Reviewer #4, Question 5. Minor comment. The discussion section should briefly explain how the saline-induced hyperchloremia can influence renal function.

Authors’ response, Question 5. Thank you for this excellent comment and suggestion. In the discussion we have now briefly outlined the proposed mechanism of hyperchloraemia and adverse renal function. We state that “Numerous studies show that changes in serum chloride concentration, independent of serum sodium and bicarbonate, are associated with increased risk of AKI. Hyperchloraemia-associated AKI is thought to be related to renal vasoconstriction mediated by tubulo-glomerular feedback, and possibly other mechanisms, given the role of chloride in regulating renal blood flow, glomerular filtration rate, tubulo-glomerular feedback, and tubular injury.”

Reference: Rein JL, Coca SG. "I don't get no respect": the role of chloride in acute kidney injury. Am J Physiol Renal Physiol. 2019 Mar 1;316(3):F587-F605. doi: 10.1152/ajprenal.00130.2018. Epub 2018 Dec 12. PMID: 30539650; PMCID: PMC6459301.

Once again, thank you for taking the time to review and consider our manuscript for publication in PLOS ONE. The comprehensive reviews and constructive comments provided by yourself and the expert reviewers have been immensely appreciated. 

A/Prof Laurence Weinberg

BSc, MBBCh,MRCP,DPCritCareEcho,FANZCA,MD

Director, Department of Anaesthesia, Austin Hospital

Associate Professor, Department of Surgery, University of Melbourne

Associate Professor, Perioperative Pain and Medicine Unit, Department of Surgery, University of Melbourne

---

## [Decision Letter · Decision Letter 1]

15 Mar 2021

PONE-D-20-37578R1

The effects of 0.9% saline versus Plasma-Lyte 148 on renal function as assessed by creatinine concentration in patients undergoing major surgery: a single-centre double-blinded cluster crossover trial

PLOS ONE

Dear Dr. Weinberg,

Thank you for submitting your manuscript to PLOS ONE. After careful consideration, we feel that it has merit but does not fully meet PLOS ONE’s publication criteria as it currently stands. Therefore, we invite you to submit a revised version of the manuscript that addresses the points raised during the review process.

As authors can see one of the reviewers still have some major concerns that I do agree. 

We look forward to receiving your revised manuscript.

Kind regards,

Felipe Dal Pizzol

Academic Editor

PLOS ONE

Reviewers' comments:

Reviewer's Responses to Questions

**Comments to the Author**

1. If the authors have adequately addressed your comments raised in a previous round of review and you feel that this manuscript is now acceptable for publication, you may indicate that here to bypass the “Comments to the Author” section, enter your conflict of interest statement in the “Confidential to Editor” section, and submit your "Accept" recommendation.

Reviewer #1: All comments have been addressed

Reviewer #2: (No Response)

Reviewer #4: (No Response)

2. Is the manuscript technically sound, and do the data support the conclusions?

Reviewer #1: Yes

Reviewer #2: (No Response)

Reviewer #4: Partly

3. Has the statistical analysis been performed appropriately and rigorously? 

Reviewer #1: Yes

Reviewer #2: (No Response)

Reviewer #4: Yes

4. Have the authors made all data underlying the findings in their manuscript fully available?

Reviewer #1: Yes

Reviewer #2: (No Response)

Reviewer #4: No

5. Is the manuscript presented in an intelligible fashion and written in standard English?

Reviewer #1: Yes

Reviewer #2: (No Response)

Reviewer #4: Yes

6. Review Comments to the Author

Reviewer #1: All comments have been addressed. I have no further comments. I commend the authors for their work.

Reviewer #2: COMMENTS: All the comments made on earlier draft(s) by me (and hopefully by other respected reviewers also) were/are attended positively/adequately, I am fully satisfied and the manuscript is improved a lot. Excellent job. I recommend acceptance.

Quoted article on ‘Understanding the cluster randomised crossover design: a graphical illustration of the components of variation and a sample size tutorial’ is excellent and new for me. Thanks a lot.

Reviewer #4: The authors do not seem to understand the basic concepts : They fail to recognize the basic mechanism by which saline solutions may alter the renal function, i.e., the hyperchloremia, with the secondary metabolic acidosis. Saline-treated patients had peak serum chloride levels at 103 [100 to 106] mmol/l vs 101 [98 to 103] mmol/L), there is no reason to believe in any difference in renal function. The conclusions of the present study are that the administration of 2 L of either saline or Plasmalyte does not affect the chloride levels and therefore cannot result in any difference in renal function, as assessed by the creatinine concentration. That is interesting.

In the study by Stemler et al (N Engl J Med 2018), there were more MAKE in the saline treated patients, but their chloride levels increased to a mean of 112 mmol/L.

For explanations, to can refer to Vincent and DeBacker ‘we do not appreciate SALT’ in the AmJRespCrit Care Med 2017.

A figure showing chloride levels over time is missing, when this is the core of the paper. This information should be included in the conclusions of the paper. Even pragmatic clinical trials must include a sound scientific rationale.

7. PLOS authors have the option to publish the peer review history of their article (what does this mean?). If published, this will include your full peer review and any attached files.

Reviewer #1: **Yes: **Fernando Zampieri

Reviewer #2: **Yes: **Dr. Sanjeev Sarmukaddam

Reviewer #4: **Yes: **JL Vincent

---

## [Author Response · Author response to Decision Letter 1]

7 Apr 2021

PONE-D-20-37578R1

The effects of 0.9% saline versus Plasma-Lyte 148 on renal function as assessed by creatinine concentrations in patients undergoing major surgery: a single-centre double-blinded cluster crossover trial

On behalf of my co-authors, I would like to sincerely thank each Reviewer for considering our revised manuscript for publication. The comprehensive reviews and constructive comments provided by yourself and the expert reviewers have been immensely appreciated. The insightful and constructive comments have enabled us to strengthen the scientific rigor and merits of our research. Thank you once again for considering our resubmission for publication. 

Reviewer 1: Professor Fernando Zampieri 

Reviewer #1 All comments have been addressed. I have no further comments. I commend the authors for their work.

Authors’ reply. Thank you once again for taking the time to review our resubmission manuscript and for your positive comments. 

Reviewer 2: Dr Sanjeev Sarmukaddam 

Reviewer 2: All the comments made on earlier drafts by me (and hopefully by other respected reviewers also) were/are attended positively/adequately, I am fully satisfied, and the manuscript is improved a lot. Excellent job. I recommend acceptance. Quoted article on ‘Understanding the cluster randomised crossover design: a graphical illustration of the components of variation and a sample size tutorial’ is excellent and new for me. Thanks a lot.

Authors’ reply. Thank you Dr Sarmukaddam for your expert comments and for taking the time to review our revised manuscript. 

Reviewer 4: Professor JL Vincent 

Reviewer 4: The authors do not seem to understand the basic concepts. They fail to recognize the basic mechanism by which saline solutions may alter the renal function, i.e., the hyperchloremia, with the secondary metabolic acidosis. Saline-treated patients had peak serum chloride levels at 103 [100 to 106] mmol/l vs 101 [98 to 103] mmol/L), there is no reason to believe in any difference in renal function. The conclusions of the present study are that the administration of 2 L of either saline or Plasmalyte does not affect the chloride levels and therefore cannot result in any difference in renal function, as assessed by the creatinine concentration. That is interesting.

In the study by Stemler et al (N Engl J Med 2018), there were more MAKE in the saline treated patients, but their chloride levels increased to a mean of 112 mmol/L. For explanations, to can refer to Vincent and DeBacker ‘we do not appreciate SALT’ in the Am J Resp Crit Care Med 2017.

A figure showing chloride levels over time is missing, when this is the core of the paper. This information should be included in the conclusions of the paper. Even pragmatic clinical trials must include a sound scientific rationale.

Question 1. Authors’ response: Thank you once again for your constructive comments. Once again, we feel privileged to have such an internationally distinguished academic clinician review our manuscript. We are genuinely grateful for your advice regarding further modifications to our manuscript. 

We are very familiar with the excellent paper by Semler and we have already made reference to this in our manuscript. The article by Vincent JL, De Backer D. We Do Not Appreciate SALT. Am J Respir Crit Care Med. 2018 May 15;197(10):1361 is also well known to our group. We acknowledge many of the important issues regarding hyperchloraemia that were raised in this thought-provoking Editorial. We agree with Professor Vincent, as stated in the Editorial, that further information discussing the potential harmful effects of hyperchloraemic solutions would strengthen our manuscript. 

We have now included the following addition information in the discussion section of the revised manuscript. We have also included four more references supporting our discussion. 

In the third paragraph of our discussion, we state now state “Our findings imply that crossover design studies investigating fluid therapy in the perioperative setting are feasible. Moreover, our findings suggest that clinicians can reasonably use either solution intraoperatively for adult patients undergoing major emergency or elective procedures if the volume of fluid administered is less than 2000 mL. However, concerns about the potentially harmful effects of hyperchloraemia remain. The clinical practice of using IV fluid solutions with unphysiological concentrations of chloride has been strongly discouraged due to the significant association between hyperchloraemia and the development of metabolic acidosis and adverse renal outcomes [30-32]. Moreover, for clinical trials investigating the effects of hyperchloraemia versus balanced solutions, clinical equipoise has also been recently questioned for the continued use of hyperchloremic solutions. This is particularly true if patients who develop hyperchloraemia continue to receive solutions with supraphysiological concentrations of chloride [30]. Given that numerous studies have shown a harm signal associated with hyperchloraemic crystalloid solutions, without a comparable benefit signal [33], the ethical integrity of clinical trials comparing saline to balanced solutions has also come under scrutiny [30].

In the present study, we acknowledge that the low chloride load associated with the administration of small volumes of saline was well tolerated by our patient cohort, who were at low risk for AKI. Moreover, we also acknowledge that our findings are not generalisable to patients with pre-existing renal dysfunction, those with metabolic acidosis and those receiving large volumes of hyperchloraemic solutions [31]. Recent editorials have emphasised that renal function is frequently unknown during surgery, and the volume of fluid replacement is unpredictable in acute care settings [31]. Therefore, because there is no evidence of benefit from hyperchloraemic solutions (other than in the management of metabolic alkalosis or hyponatraemia), and there is evidence of potential harm, there appears to be no physiologic or clinical justification for the continued use of “unphysiological” saline (or any other unbalanced crystalloid solution) for perioperative volume therapy, especially when safer and more physiological solutions are available [31,32,33]. 

Additional references

30. Vincent JL, De Backer D. We Do Not Appreciate SALT. Am J Respir Crit Care Med. 2018 May 15;197(10):1361. DOI: 10.1164/rccm.201709-1874LE. 

31. Priebe HJ. Another nail in the saline coffin. Br J Anaesth. 2018 Jun;120(6):1432-1434. DOI: 10.1016/j.bja.2018.02.026. 

32. Kellum JA, Shaw AD. Assessing toxicity of intravenous crystalloids in critically ill patients. JAMA 2015 Oct 27;314(16):1695-7. DOI: 10.1001/jama.2015.12390.

33. McLean DJ, Shaw AD. Intravenous fluids: effects on renal outcomes. Br J Anaesth. 2018 Feb;120(2):397-402. DOI: 10.1016/j.bja.2017.11.090. 

Question 2. Authors’ response: In addition, we have now included another Figure showing chloride levels over time. The addition of Figure 3 graphically outlines the preoperative and postoperative changes in chloride levels. We have provided violin plots rather than the traditional box plots. We think that violin plots are more informative than a plain box plot, which only shows summary statistics such as median and interquartile ranges. In contrast, the violin plot shows the full distribution of the data. The difference is particularly useful when the data distribution is multimodal (more than one peak). By using violin plots, we hope to demonstrate the presence of different chloride peaks, their position and relative frequency.

Figure 3. Preoperative and postoperative changes in chloride levels. The single thicker dashed line represents the median value, and the thinner dashed lines represent the interquartile ranges. 

Question 3. We have also deleted the sentence in the conclusion stating that “Our findings also imply that clinicians can reasonably use either solution intraoperatively for adult patients undergoing major surgery.”

In summary, we hope we have adequately addressed the constructive comments by Professor Vincent and would be very happy to make further modifications if requested. Once again, we thank Professor Vincent for taking the time to review and consider our manuscript for publication in PLOS ONE. His comprehensive review and constructive comments provided, in addition to the positive comments from the other reviewers during the first resubmission have been immensely appreciated. 

Sincerely, 

A/Prof Laurence Weinberg

BSc, MBBCh,MRCP,DPCritCareEcho,FANZCA,MD

Director, Department of Anaesthesia, Austin Hospital

Associate Professor, Department of Surgery, University of Melbourne

Associate Professor, Perioperative Pain and Medicine Unit, Department of Surgery, University of Melbourne

---

## [Decision Letter · Decision Letter 2]

3 May 2021

The effects of 0.9% saline versus Plasma-Lyte 148 on renal function as assessed by creatinine concentration in patients undergoing major surgery: a single-centre double-blinded cluster crossover trial

PONE-D-20-37578R2

Dear Dr. Weinberg,

We’re pleased to inform you that your manuscript has been judged scientifically suitable for publication and will be formally accepted for publication once it meets all outstanding technical requirements.

Kind regards,

Felipe Dal Pizzol

Academic Editor

PLOS ONE

Additional Editor Comments (optional):

Reviewers' comments:

Reviewer's Responses to Questions

**Comments to the Author**

1. If the authors have adequately addressed your comments raised in a previous round of review and you feel that this manuscript is now acceptable for publication, you may indicate that here to bypass the “Comments to the Author” section, enter your conflict of interest statement in the “Confidential to Editor” section, and submit your "Accept" recommendation.

Reviewer #4: All comments have been addressed

2. Is the manuscript technically sound, and do the data support the conclusions?

Reviewer #4: Yes

3. Has the statistical analysis been performed appropriately and rigorously? 

Reviewer #4: Yes

4. Have the authors made all data underlying the findings in their manuscript fully available?

Reviewer #4: Yes

5. Is the manuscript presented in an intelligible fashion and written in standard English?

Reviewer #4: Yes

6. Review Comments to the Author

Reviewer #4: the paper is much better

the discussion is sound and the Figure 3 excellent

the message is much clearer

7. PLOS authors have the option to publish the peer review history of their article (what does this mean?). If published, this will include your full peer review and any attached files.

Reviewer #4: **Yes: **Jean-Louis Vincent

---

## [Editor Report · Acceptance letter]

10 May 2021

PONE-D-20-37578R2 

The effects of 0.9% saline versus Plasma-Lyte 148 on renal function as assessed by creatinine concentration in patients undergoing major surgery: a single-centre double-blinded cluster crossover trial 

Dear Dr. Weinberg:

I'm pleased to inform you that your manuscript has been deemed suitable for publication in PLOS ONE. Congratulations! Your manuscript is now with our production department. 

Kind regards, 

on behalf of

Dr. Felipe Dal Pizzol 

Academic Editor

PLOS ONE